# A universal surface functionalization technique to chemically enhance live microbial cells

Gabriel T Vercelli [1], Xingcheng Zhou[2], Stefany Moreno-Gámez[1], Rashi R Jeeda[3], Rachel Gregor [4], Jonasz Słomka [5], Akorfa Dagadu[1], Ariel L Furst [2✉] & Otto X Cordero [1✉]

## Abstract

**Microbial surface functionalization is a powerful strategy for endowing microbes with novel, non-genetic functions. However, existing methods are often species-specific, limited in scope, and compromise cell viability. Here, we present a universal and modular platform for high-density, reproducible surface functionalization across diverse microbial species—including Gram-positive, Gram-negative, aerobic, and anaerobic bacteria—using multiple molecular classes such as fluorophores, enzymes, and nucleic acids. Our method preserves cell viability and achieves 50× higher functionalization efficiency than previous methods with a standardized protocol applicable to any azide-containing molecule. Applications of the method show reproducible and tunable phenotypic outcomes at the single-cell level: fluorophore labeling yielded adjustable fluorescence, β-lactamase conferred scalable antibiotic resistance, and DNA coatings modulated adhesion and aggregation. This platform provides quantitative, non-genetic control over microbial phenotypes and complements genetic engineering approaches. It enables new possibilities for microbial design in biotechnology, medicine, and environmental applications where genetic modification is impractical or undesirable.**

**Keywords** Cell Surface Functionalization; Click Chemistry; Non-genetic Engineering; Microbial Aggregates; Molecular Engineering
**Subject Categories** Methods & Resources; Microbiology, Virology & Host Pathogen Interaction

## Introduction

Bacteria are essential in many natural and industrial processes, ranging from biogeochemical cycling to waste treatment, food production, and human health. In all these processes, bacteria interact with their environment through their cell surface, which harbors receptors for biochemical signaling and mediates molecular transfer between the extracellular and intracellular environments. Because of its importance, the bacterial cell surface has been a major target for bioengineering (Wu and Liu, 2022; Liu et al, 2019; Lin et al, 2023; Han et al, 2024) with applications including enhanced gut adhesion (Vargason et al, 2020; Vargason and Anselmo, 2020; Song et al, 2022) and resistance to environmental stressors for probiotics (Pan et al, 2022, 2021), improved drug delivery and tumor diagnosis (Moreno et al, 2020; Lahav-Mankovski et al, 2020; Wu and Liu, 2022), efficient removal of environmental pollutants (Pan et al, 2019; Feng et al, 2023), and facile manufacturing of microbial fuel cells (Furst et al, 2018).

Despite the clear advantages of surface-functionalized cells for a variety of applications, several technical limitations have prevented their widespread adoption and clinical translation (Lin et al, 2023; Liu et al, 2019; Han et al, 2024). Broadly speaking, there are three classes of methods used to modify bacterial cell surfaces: those based on genetic engineering, metabolic labeling, and chemical modification. Genetic engineering-based methods have the advantage of permanently maintaining new functions in cells, since new functional molecules are constantly being produced. However, these methods are limited by the narrow range of genetically tractable bacteria and by concerns associated with the use of genetically modified organisms in the open environment (Kim et al, 2022, 2020; Glass and Riedel-Kruse, 2018; Gonçalves and Paiva, 2017; Arnold et al, 2023). Metabolic labeling methods, which rely on the uptake of chemical analogs of metabolites, have been a great tool to study a cell's native metabolism, allowing the spatial visualization of cellular functions and unprecedented quantitation of metabolic rates. Nevertheless, as a consequence of their high dependence on metabolism, these methods show variable efficiencies across microbes, can disrupt the native metabolic pathways of the cell, and require long preparation times for slow growing organisms (Siegrist et al, 2015; van Kasteren and Rozen, 2023; Hajjo et al, 2023; Shalizi et al, 2020; Landor et al, 2023). Lastly, direct chemical modification methods are fast and highly versatile, but often require synthesis of specialized compounds and extensive optimization of reaction conditions for each application due to unpredictable chemical toxicity effects (Twite et al, 2012; Xie et al, 2017; Geng et al, 2021; Tian et al, 2020; Liu et al, 2019; Lin et al, 2023). Beyond the specific advantages and disadvantages of all these

[1]Department of Civil and Environmental Engineering, Massachusetts Institute of Technology, Cambridge, MA, USA. [2]Department of Chemical Engineering, Massachusetts Institute of Technology, Cambridge, MA, USA. [3]Department of Biological Engineering, Massachusetts Institute of Technology, Cambridge, MA, USA. [4]Department of Chemical Engineering and Applied Chemistry, University of Toronto, Toronto, ON, Canada. [5]Department of Civil, Environmental and Geomatic Engineering, Institute of Environmental Engineering, ETH Zurich, Zurich, Switzerland. ✉E-mail: afurst@mit.edu; ottox@mit.edu

methods, a major limitation of the field is the lack of standardized assays and systematic studies quantitatively comparing existing surface functionalization methods. This lack of standardization has made it challenging to compare the trade-offs of each method and slowed progress in the field (Wang et al, 2022).

In this work, we pursue two goals: (1) to introduce a novel surface functionalization technique that overcomes key limitations of existing methods and efficiently labels bacterial cell surfaces while maintaining high viability, and (2) to establish standardized, quantitative assays for evaluating functionalization performance. To address the first goal, we developed a new direct chemical modification technique based on hetero-bifunctional crosslinkers— molecules containing two orthogonal reactive groups—that enable modular surface labeling by decoupling the species-specific protocol optimization step from the choice of added functionality. The use of such molecules is prevalent in the biomolecule-drug and biomolecule-exosome conjugation space due to the versatility they provide (Tian et al, 2018; Zhang et al, 2024; Brinkley, 1992). However, due to the challenges of adapting crosslinking reaction conditions to biological constraints, applications to live cell surface functionalization remain underexplored, with biotinylation of cell surfaces through N-hydroxysulfosuccinimide (sulfo–NHS) ester chemistry being the only well-established biocompatible technique (Vargason and Anselmo, 2020; Vargason et al, 2020; He et al, 2023). Our approach expands this toolkit, enabling efficient and non-toxic conjugation of azide-containing molecules to the cell surface. We then turned to our second goal, using standardized assays to benchmark our method against prior techniques across three key criteria: cell viability, functionalization efficiency, and protocol duration. Our results show clear advantages on all fronts, establishing the robustness of this technique.

To demonstrate the versatility and impact of our method, we applied it to a broad range of functionally relevant and taxonomically diverse microbes, including polymer-degrading marine isolates and well-established probiotic strains. We then focused on enhancing these cells with three core capabilities shown to have significant applications in biomedicine and bioengineering: fluorescent labeling, antibiotic resistance, and programmable adhesion. To achieve these new functions, we conjugated cells with fluorophores, beta-lactamase enzymes, and single-stranded DNA, respectively, additionally showcasing the compatibility of our method with a diversity of molecules. Each of these functionalization efforts yielded strong and tunable acquisition of the expected phenotype. Together, these results demonstrate how our technique enables fast, efficient, and broadly compatible surface modification of cells and lay the foundation for scalable microbial engineering strategies that were previously inaccessible.

## Results

### Covalent functionalization of bacterial cell surfaces with azide-containing molecules

Bacterial cell surfaces naturally display a variety of chemical groups that can serve as targets for covalent functionalization (Wu and Liu, 2022; Liu et al, 2019; Lin et al, 2023). Each of these native residues can be modified with specific chemical methods. In this work, we focused on modifying free primary amines, which are abundant

across microbial species and commonly found at the N-terminus of surface proteins and in lysine side chains (Seltmann and Holst, 2002; Perkins, 2012). These groups can be selectively modified using N-hydroxysuccinimide (NHS) ester chemistry, but optimal reaction conditions that are also biocompatible are not clearly defined (Mädler et al, 2009). Therefore, we devised a surface functionalization strategy that uses dibenzocyclooctyne-sulfo-NHS (DBCO–sulfo–NHS), a hetero-bifunctional crosslinker, to covalently functionalize the cell surface and performed screens to find optimal reaction conditions. The presence of the sulfo group in the NHS moiety enhances the hydrophilicity of the crosslinker, eliminating the need for organic solvents and reducing cellular internalization, an advantage that we hypothesized would help maintain high cell viability. Moreover, upon NHS-mediated functionalization of surface amines with this crosslinker, DBCO moieties become exposed on the cell surface, allowing subsequent rapid and specific conjugation of azide-containing molecules via strain-promoted azide-alkyne cycloaddition (SPAAC) (Sletten and Bertozzi, 2009). Therefore, this two-step strategy enables efficient and modular surface functionalization with any azide-labeled molecule (Fig. 1).

To validate this surface functionalization strategy, we modified the surface of Escherichia coli cells with a high density of azide-modified fluorophores (Fig. 2). After functionalization, cells were imaged with confocal microscopy. Cells treated with DBCO–sulfo–NHS and Alexa Fluor 488 azide—the chosen fluorophore—showed bright green fluorescence, indicating successful surface functionalization, while controls in which the linker or fluorophore was not added showed no background fluorescence, confirming that non-specific binding of the azide-fluorophore and cellular auto-fluorescence were not an issue (Fig. 2A; Appendix Fig. S1a–d). Thus, the cell functionalization with this bifunctional crosslinker demonstrates specificity for reactivity with azides.

Although these results qualitatively show successful surface functionalization, approaches to chemically modify cells often result in a trade-off between modification efficiency and cell viability. Therefore, we aimed to quantitatively characterize the modification efficiency and cell viability induced by our method as the concentration of our chemical treatments increased. To achieve this goal experimentally, cells were treated with a range of crosslinkers (0–1000 mg/L) and azide-modified fluorophore (0.4–40 μM) concentrations. The effect on viability was measured by subsequently staining cells with the dead cell strain SytoX Orange, following standard protocols. These samples were then analyzed by flow cytometry, where green and red single-cell fluorescence were measured as proxies for modification efficiency and cell viability, respectively. To convert our efficiency measurements from fluorescence units to the number of molecules loaded onto the cell surface, we constructed calibration curves using fluorescent beads (Appendix Fig. S2a,b; "Methods"). For calculating viability, a simple fluorescence threshold separating labeled from non-labeled populations was used (details in "Methods").

An increase in green fluorescence was observed as either the crosslinker or azide probe concentrations were increased (Fig. 2B; Appendix Fig. S2c), indicating that the degree of cell modification can be controlled based on the chemical concentrations used. At very high crosslinker concentrations, increases in green fluorescence were accompanied by slightly higher proportions of red-stained cells (Fig. 2C), indicating increased cell death under these

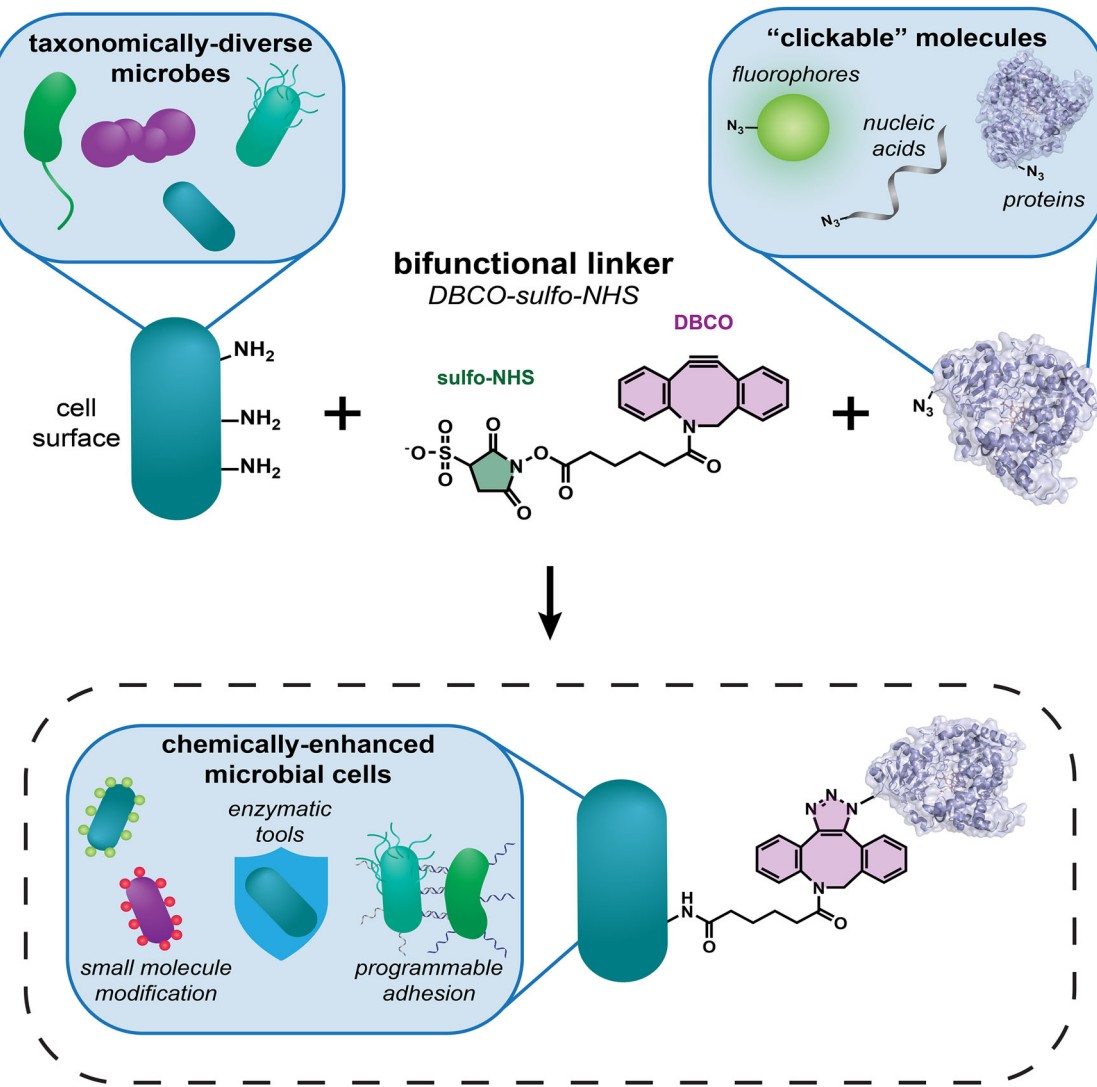

**Figure 1.  General strategy for surface functionalization of microbial cells in two steps.**

(1) Microbial cells are incubated with DBCO–sulfo–NHS, an amine-reactive bifunctional linker molecule. The sulfo–NHS group (green) reacts with naturally occurring free amines on the cell surface. (2) The DBCO group (purple) is then used in a SPAAC reaction to couple azide-modified molecules to the microbial cell surface. The resulting microbial cells gain novel functions like fluorescent labeling, an extended enzymatic repertoire, and programmable adhesion. Source data are available online for this figure.

conditions. Conversely, high concentrations of the azide probe showed no changes in cell viability (Appendix Fig. S2c). We therefore tested incubation conditions with higher azide probe concentrations (400 μM) and higher temperatures (37 °C). Neither of these changes led to significant increases in fluorescence, confirming that the DBCO-azide reaction is close to saturation (Appendix Fig. S2d,e). These results confirm that while there is a trade-off between modification efficiency and viability at high concentrations, at a wide range of working concentrations, the linker has no negative effect on cell viability (Fig. 2E).

Although membrane integrity is an accepted marker of viability for microbes, it does not take into account lag times in the growth of cells that may be induced by the treatment. Alternative methods of viability estimation like colony-forming unit (CFU) counting also do not take these growth delays into account. Since cellular activity is the most important trait for downstream applications of

functionalized cells, we developed a new assay that estimates the proportion of active cells in the population by measuring lag times in growth. We call it the regrowth dynamics assay, as it uses the regrowth dynamics of the cells following treatment to directly measure cell activity. This assay is similar to the well-established start growth time method (SGT) (Hazan et al, 2012) and standardizes how regrowth data should be analyzed and reported. In short, we quantified the regrowth dynamics of treated and untreated cultures by measuring optical density (OD) (Fig. 2D). We then fitted the optical density data to an exponential growth model for strains grown in minimal media, and to a logistic growth model for strains grown in rich media. From these curve fits, we estimated the initial active population ($N_0$) and the population growth rate ($r$) in each condition. As we did not explicitly fit a lag time to the model, the initial active population estimation decreases with any potential extra lag time in treatment conditions, resulting in lower

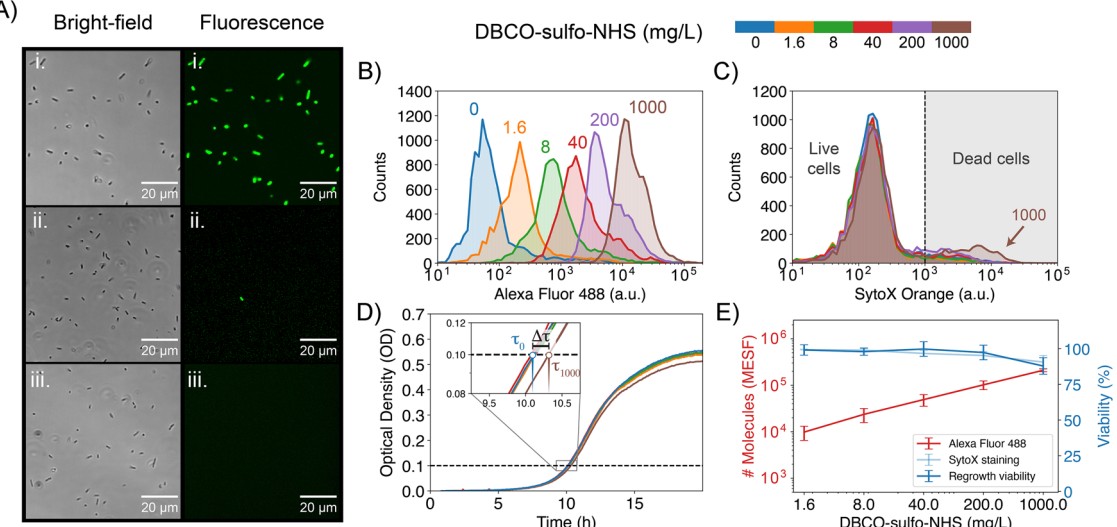

**Figure 2. Validation of surface modification protocol for *E. coli* by characterizing the viability-efficiency trade-off of the technique.**

(A) Bright-field and fluorescence microscopy images showing *E. coli* MG1655 cells treated with (i) DBCO–sulfo–NHS and Alexa Fluor 488 azide, (ii) Alexa Fluor 488 azide only, and (iii) untreated cells. Only fully treated cells (i) acquire bright green fluorescence. Scale bars are shown in the images. (B, C) Flow cytometry quantification of green (B) and red (C) fluorescence intensity in arbitrary units (a.u.) of *E. coli* cells treated with increasing concentrations of DBCO–sulfo–NHS (0–1000 mg/L) followed by Alexa Fluor 488 azide, and the dead cell stain SytoX Orange. Increasing the DBCO–sulfo–NHS concentration proportionately increases the intensity of the fluorescence labeling (B), with minimal increase in stained dead cells (C). (D) Growth curves for *E. coli* treated with increasing concentrations of DBCO–sulfo–NHS (0–1000 mg/L). The inset depicts a zoom-in of the curves in log-scale showing increasing delays in growth with increased DBCO–sulfo–NHS treatment strength but no changes in growth rate (slope). Lag time in growth () between 0 and 1000 mg/L treatment is depicted in the image. (E) Effect of increasing DBCO–sulfo–NHS concentrations on: (1) number of Alexa Fluor 488 azide molecules tethered to the surface (red, left axis), estimated in Molecules of Equivalent Soluble Fluorochrome (MESF) units using calibration curves constructed with fluorescent beads, and (2) cell viability, calculated from SytoX Orange staining (light blue, right axis) and from regrowth method (dark blue, right axis). The error bars for the number of tethered molecules represent SEM across three biological replicates, and for the viability SEM across four biological replicates. Source data are available online for this figure.

viability estimates. We then calculated the effect of the treatment on cell activity as the ratio of the active population in a treated sample to that in the untreated control (details in "Online Methods"). Overall, this calculation is more stringent than standard viability measures, since viable cells with slower regrowth dynamics contribute less than untreated cells to our estimation of cell activity.

The viability results obtained with the regrowth method for *E. coli* cells closely match the viability estimates calculated from cell permeability dyes (Fig. 2E). For instance, at the highest concentration of the linker molecule, 1000 mg/L, we estimate $(88 \pm 6)\%$ viability with the regrowth assay and $(91 \pm 5)\%$ viability with cell permeability dyes. For both methods, the estimated cell viability in all treatment conditions did not significantly differ from 100% (one-sided Student $t$ test, Fig. EV1 and "Methods"). In addition, in all conditions the population growth rate did not significantly differ from the control (one-sided Student $t$ test, Appendix Fig. S3 and "Methods"). As a consequence, the highest treatment condition considered, 1000 mg/L, was chosen for follow-up experiments with *E. coli*. In this condition, 210,000 molecules are loaded per cell, which translates to a high surface density of approximately one molecule every 30 nm² or a mean distance of about 5 nm between adjacent molecules. To estimate the loss of surface modification with growth, *E. coli* cells were functionalized with fluorophores and imaged with a fluorescence microscope over a couple of divisions. Single lineage fluorescence was then quantified and fitted against the number of doublings showing that fluorescence strength decayed by half with each area doubling (Fig. EV2).

In addition to functionalizing *E. coli* cells, we also tested whether our method could be applied to a taxonomically diverse panel of five additional microbial isolates (Fig. 3). We included both Gram-positive and Gram-negative strains to test how their distinct cell surface organization affected the efficiency of the technique. We evaluated two environmental strains isolated from seawater, the Gram-negative, aerobic bacteria *Vibrio splendidus* 1A01 and *Neptunomonas sp.* 3B05 (Datta et al, 2016); the Gram-negative, anaerobic type strain *Bacteroides ovatus* ATCC 8483, which is part of the Human Microbiome Project collection; and two Gram-positive, anaerobic probiotic species isolated from human stool, *Lactococcus lactis* RJX1236 and *Lactobacillus rhamnosus* RJX1228 (Vatanen et al, 2016). Since all three gut isolates are cultured primarily in anaerobic conditions, we performed our functionalization protocol anaerobically for those strains. As with our *E. coli* experiments, we treated cells with a range of crosslinker concentrations and incubated them with Alexa Fluor 488 azide. Flow cytometry was then used to quantify the number of fluorescent molecules attached to the cell surface (Appendix Fig. S4). Viability was estimated with the regrowth dynamics method since it is more stringent than the permeability dye staining and requires less strain-specific fine-tuning (Appendix Fig. S5). Based on these measurements, we built trade-off curves mapping out the effects of crosslinker concentration on functionalization efficiency and cell viability (Figs. 3A and EV1) and for each species identified treatment conditions that did not have a significant impact on viability (one-sided Student $t$ test, "Methods") and still

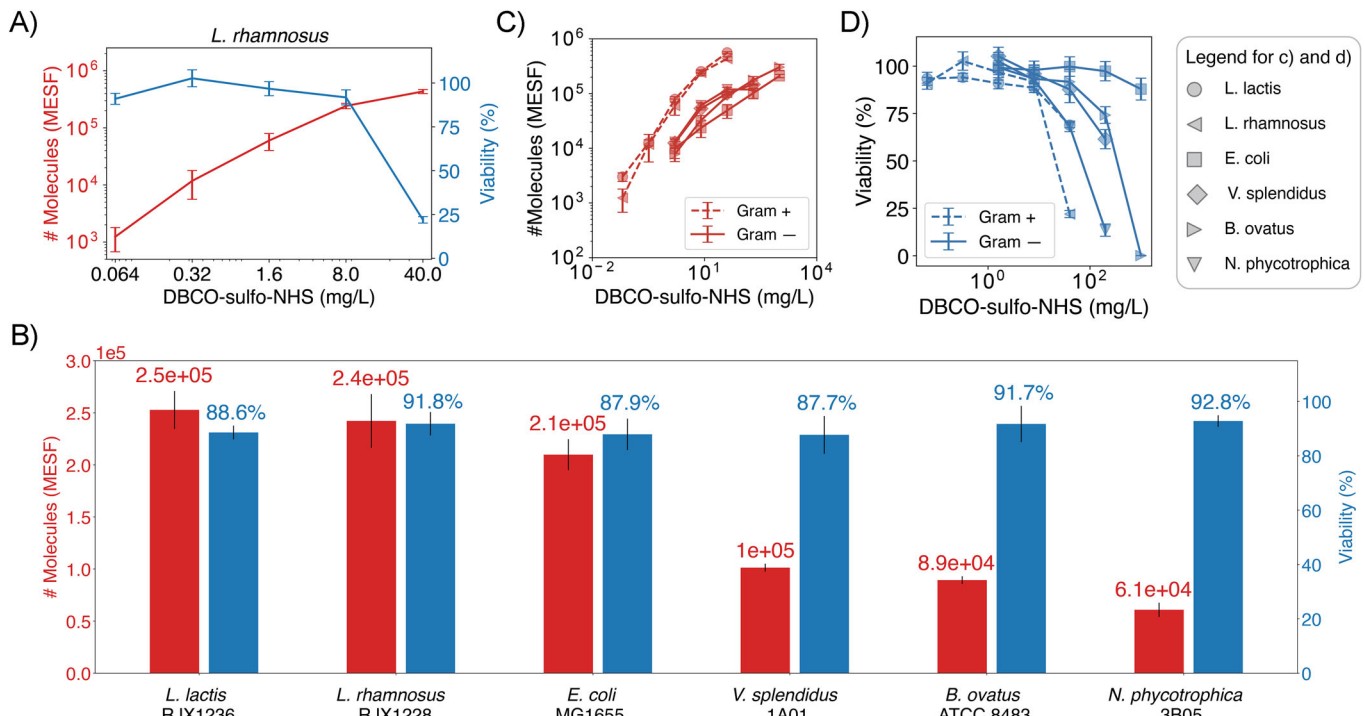

**Figure 3. Validation of surface modification protocol for five additional taxonomically diverse microbial species by characterizing the viability-efficiency trade-off of the technique.**

(**A**) Trade-off quantification for *L. rhamnosus* strain, a Gram-positive, anaerobic microbe, depicting the effect of increasing DBCO–sulfo–NHS concentrations on: (1) number of Alexa Fluor 488 azide molecules tethered to the surface in MESF units (red, left axis) and (2) cell viability calculated from the regrowth method (blue, right axis). (**B**) Number of surface tethered molecules and cell viability at crosslinker concentrations that maintain cell viability above 80%. Surface functionalization levels of 60,000–250,000 surface tethered molecules per cell are achieved across all species. (**C**, **D**) Number of Alexa Fluor 488 azide molecules tethered to the cell surface (**C**) and cell viability (**D**) as a function of DBCO–sulfo–NHS concentration for the six strains screened. Gram-positive strains (circle and left-triangle) acquire more surface molecules than Gram-negative strains (square, diamond, right and down-triangles) for the same DBCO–sulfo–NHS concentration. However, Gram-positive strains show viability losses at lower crosslinker concentrations than Gram-negative ones. In all panels, viability error bars represent SEM across four biological replicates for *L. lactis*, *L. rhamnosus*, and *E. coli*, and three biological replicates for all other strains. The number of molecules error bars represent SEM across two biological replicates for *L. rhamnosus* and three biological replicates for all other strains. Source data are available online for this figure.

tethered 60,000–250,000 molecules onto the surface of single cells (Fig. 3B). In addition, we showed that there were no significant impacts to growth rate in these conditions (Appendix Fig. S3). This corroborates the compatibility of the method with Gram-positive and Gram-negative natural isolates in aerobic and anaerobic conditions and highlights the potential of the technique to engineer natural microbial communities.

Further comparing Gram-positive and Gram-negative bacteria, the number of surface tethered molecules per cell was reproducible within each Gram-staining group, but large differences were seen between them (Fig. 3C). Gram-positive strains were more prone to surface modification, showing higher numbers of surface tethered molecules than the ones for Gram-negative strains exposed to the same crosslinker treatment concentration. Regarding cell viability, although strain sensitivity to the crosslinker treatment was highly variable, Gram-positive strains were consistently more sensitive to DBCO–sulfo–NHS treatment than Gram-negative bacteria, showing higher viability losses when treated with the same crosslinker concentration (Fig. 3D). These results highlight the challenges of developing a surface functionalization technique compatible with taxonomically diverse strains and support the universality of our approach.

Benchmarking these results against previous methods, we see that our approach can achieve higher surface functionalization efficiencies while maintaining high cell viability and short protocol times (Table 1). Specifically, metabolic labeling of *E. coli* cells with azidohomoalanine (AHA) (Link and Tirrell, 2003) or with β-D-glucopyranosyl azide (Kong et al, 2025) achieved at most a tenfold increase in mean fluorescence intensity (MFI) of cells, while our method can achieve a 110-fold increase. Sodium periodate oxidation of surface aldehydes followed by modification with hydrazines through a Wolff-Kishner reduction reduced the viability of *E. coli* to less than 0.01% (Appendix Fig. S6a–d) and is reported to functionalize *A. vinelandii* with 26,000 (Twite et al, 2012), 8 times lower than what our method achieves in *E. coli* and three times lower than what we can achieve in any other strain assayed. Surface functionalization of *E. coli* with biotin–sulfo–NHS crosslinkers is reported to attach less than 4000 molecules to the cell (Vargason et al, 2020). However, using the same crosslinker and the protocol developed here, we are able to achieve higher functionalization efficiencies of 15,000 molecules per cell (Table 1; Appendix Fig. S7a–c). Yet, with the DBCO–sulfo–NHS crosslinker, we achieve 210,000 functionalized molecules. This shows that the advantages of our protocol are largely related to the choice of a new

**Table 1.  Benchmarking of different surface modification methods in *E. coli*.**

| Functionalization method | Regrowth viability | Efficiency (MFI fold change) | Efficiency (MESF) | Protocol duration (h) |
|---|---|---|---|---|
| This work (DBCO–sulfo–NHS) | 88 ± 6% | 110× | 210k | 2–3 |
| EZlink with our protocol (Biotin-sulfo–NHS) | 93 ± 7% | 40× | 15k | 2–3 |
| EZlink with other protocols (Biotin-sulfo–NHS) (Vargason et al, 2020) | ~70% (estimated) | 4× (estimated) | 4k (estimated) | 1–2 |
| Hydrazine-aldehyde coupling (Twite et al, 2012) | <0.01% | 13× (*) | 26k (*) | 12–16 |
| Metabolic labeling (azide- functionalized amino acid) (Link and Tirrell, 2003) | No data | 10× | No data | 16 |
| Metabolic labeling (azide-functionalized sugar) (Kong et al, 2025) | No data | 6x | No data | 10–12 |

Comparison between cell viability, modification efficiency, and protocol duration across different surface modification methods applied to *E. coli*. The first and second rows use the protocol developed in this work. Underlined values were measured in this study, while other values were gathered from literature references. Estimated values are based on the data provided in the reference. Efficiencies are reported either as the fold change in mean fluorescence intensity (MFI) between functionalized and non-functionalized cells exposed to the same fluorophores or as molecules of equivalent soluble fluorochrome (MESF) if calibration curves were used. *These numbers were collected for *A. vinelandii* and are shown here as a comparison because numbers for *E. coli* were not available for this method. Additional metrics are compared in Table EV1.

crosslinker, but also that we were able to find better overall reaction conditions for cell surface functionalization. These results demonstrate that this new surface modification protocol can load significantly higher densities of molecules onto *E. coli* surfaces than previous techniques without major effects on cell growth or viability.

## Functional enhancement of cells with surface-bound enzymes

After determining the optimal conditions to functionalize a diverse panel of strains with fluorophores, we sought to illustrate how these same conditions generalize to other molecules. For this purpose, we chose to functionalize cells with beta-lactamase enzymes that hydrolyze beta-lactam antibiotics, rendering them inactive (Maji-duddin et al, 2002). We specifically chose these enzymes because there are standard protocols to measure their activity, making it easy to experimentally quantify the impact of our surface modification protocol on cellular function. We hypothesized that cells functionalized with these enzymes would show enhanced survival in the presence of ampicillin, a beta-lactam antibiotic, since it should be inactivated by the enzymes tethered to the microbial surface. To test this hypothesis, we treated *E. coli* cells with the previously found optimal crosslinker concentration (1000 mg/L) and incubated them with increasing concentrations of azide-modified enzymes (0, 0.4, 4, and 40 µM) produced in the lab using standard conjugation protocols (details in "Online Methods"). Then, we quantified antibiotic resistance using a minimum inhibitory concentration (MIC) assay for ampicillin (Fig. 4A).

We found that cells functionalized with beta-lactamases exhibited significant increases in MIC, showing increased survival to antibiotic treatment (Fig. 4). The effect was concentration-dependent: cells treated with 40 µM of enzyme exhibited an increase in MIC of more than 200-fold compared to non-functionalized controls, while cells treated with 4 µM of enzyme showed a twofold increase, and the ones treated with 0.4 µM of enzyme displayed no significant MIC change (Fig. 4B,C). These concentration-dependent results indicate that the antibiotic resistance is conferred by the beta-lactamase functionalization, rather than any physiological response to the chemical treatments. These

results show that the previously determined optimal functionalization conditions generalize to new molecules, demonstrating the modularity of our surface functionalization method.

We next evaluated the relative protective effects of surface-bound enzymes as compared to free enzymes in solution. We hypothesized that surface localization of enzymes could make them more efficient by spatially segregating the enzymatic function. Cells functionalized with beta-lactamases on their surface were compared to cells exposed to the same total concentration of enzymes free in solution. The concentration of enzymes in both conditions was established based on our prior measurements of fluorophore density per cell following surface modification. (Fig. 2, details in "Online Methods"). Strikingly, surface-bound enzymes achieved the same—or higher—MIC values with tenfold fewer enzymes than their free counterparts across a wide range of enzyme concentrations (Fig. 4D). This result highlights a critical advantage of enzyme localization at the surface of the microbes: tethering beta-lactamases to cell surfaces dramatically enhances the enzymes' capacity to mediate antibiotic resistance.

## Programmable cell-to-cell adhesion guided by surface-bound single-stranded DNA

Having established the potential of our surface functionalization technique to enhance the functional repertoire of individual microbial strains, we then investigated how this approach can influence multicellular communities through cell-to-cell adhesion. Intercellular interactions, including direct adhesion between different bacterial species, are critical components of native microbial communities. In particular, bacterial adhesion is involved in intra- and inter-species interactions, the formation of bacterial biofilms, and colonization of new environments (e.g., adhesion to a host's epithelial interface). However, the recapitulation and control of such interactions in synthetic cell mixtures has remained a challenge. Previous studies of mammalian and bacterial cells have shown that single-stranded DNA (ssDNA) molecules enable cell adhesion to solid surfaces in layers because of their strong and highly specific interactions with the proper complementary strand (Twite et al, 2012; Cabral et al, 2021; Todhunter et al, 2016, 2015; Hsiao et al, 2009). Inspired by these studies, we applied our

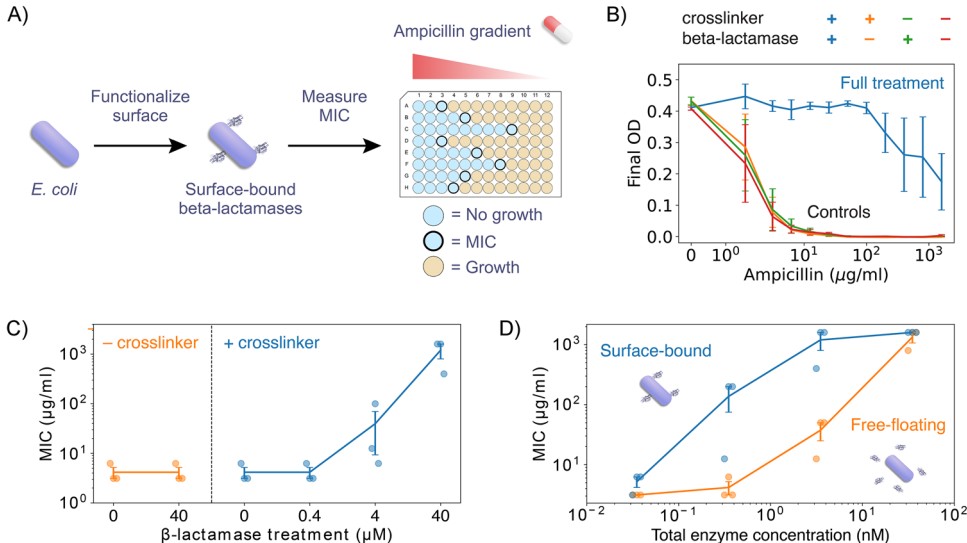

**Figure 4. Surface functionalization with beta-lactamase enzymes induces antibiotic resistance.**

(A) Schematic of experimental protocol: cells were functionalized with beta-lactamase enzymes that inactivate beta-lactam antibiotics like ampicillin. They were then grown in a gradient of ampicillin to determine the minimum inhibitory concentration (MIC) for cells with and without surface functionalization. (B) MIC assay: Final $OD_{600}$ reached by cells as a function of ampicillin concentration. Full treatment includes incubations with crosslinkers and beta-lactamases, while controls skip at least one of those steps. Only the full treatment led to effective protection against ampicillin. (C) Effect of beta-lactamase treatment concentration on MIC. Conditions are divided into two groups: with crosslinker treatment (right) and without (left). Beta-lactamase treatment concentration is shown on the $x$ axis. (D) MIC achieved by binding beta-lactamases to the cell surface compared to the one achieved by adding beta-lactamases in solution. Total enzyme concentration was matched between conditions, taking into consideration the number of surface-bound enzymes per cell and the inoculation $OD_{600}$. Surface-bound enzymes were more effective at protecting cells against ampicillin than equivalent amounts of free enzymes in solution. In all panels, error bars represent SEM of three biological replicates also shown as semi-transparent dots in (C, D). Source data are available online for this figure.

approach to create multicell assemblies of microbial strains using DNA hybridization.

To demonstrate the ability of our surface functionalization method to control bacterial adhesion, we leveraged its modularity to functionalize *E. coli* cells with azide-modified ssDNA (Fig. 5A). Using the optimized parameters established for fluorophore and enzyme attachment (Fig. 2E), we prepared two populations of cells, each functionalized with a ssDNA sequence complementary to that of the other population. Upon mixing, hybridization between complementary ssDNA strands was expected to drive cell–cell adhesion and self-assembly of the populations into dense aggregates, which we monitored using a sedimentation assay. This assay involved mixing equal numbers of both populations and measuring the optical density at a 600 nm wavelength ($OD_{600}$) of the solution over time (Fig. 5A). Aggregation of complementary ssDNA-modified cells would lead to a rapid decrease in $OD_{600}$, whereas non-complementary controls were expected to remain planktonic, maintaining a stable $OD_{600}$.

Using this sedimentation assay, we evaluated the adhesive properties of *E. coli* cells functionalized with ssDNA molecules containing 20 nucleotide bases. Just as with the enzyme functionalization experiments, we were able to tune the average amount of ssDNA tethered to the surface of each cell by changing the concentration of the ssDNA solution in which cells were incubated. Specifically, we incubated cells with 0, 0.4, 4, and 40 µM solutions of ssDNA. According to our fluorophore estimates, these conditions resulted in cells modified with 0, 15, 70, and 210k molecules on average, respectively (Appendix Fig. S2c). For each

DNA concentration, we prepared a pair of cell samples independently incubated with complementary ssDNA strands. In addition, we included a pair of samples incubated with solutions of non-complementary DNA strands at the highest concentration to validate the sequence specificity of this programmable adhesion. We then standardized the $OD_{600}$ of all samples to 1.0, mixed the samples modified with complementary DNA in cuvettes, and measured the change in $OD_{600}$ over time with a spectrophotometer (Fig. 5A).

Over the course of the experiment, measurements for the cells modified with 40 and 4 µM solutions of complementary ssDNA showed a rapid decrease in $OD_{600}$ and the formation of macroscopic aggregates after 30 min and 1 h of incubation, respectively (Fig. 5B,C). Samples modified with either 0.4 or 0 µM solutions of complementary ssDNA or with non-complementary DNA showed only a slight decrease in $OD_{600}$ and remained homogeneous (Fig. 5B,D). To extract single-cell adhesion probabilities from this population-level measurement, we fitted the initial $OD_{600}$ decrease of the samples to a simplified aggregation model (details in "Online Methods"), considering only aggregation reactions between single cells (the dominant reaction at the beginning of the experiment). From those fitting procedures, we calculated the adhesion probability per cell-to-cell encounter ($p$) for each condition tested. Our estimates show that the 0 µM and the non-complementary DNA-modified cells had a background adhesion probability of 0.20±0.01% and did not differ from each other, while the 0.4, 4, and 40 µM conditions showed 1.3-fold, 7-fold, and 15-fold increases in cell adhesion, respectively (Fig. 5E). Together,

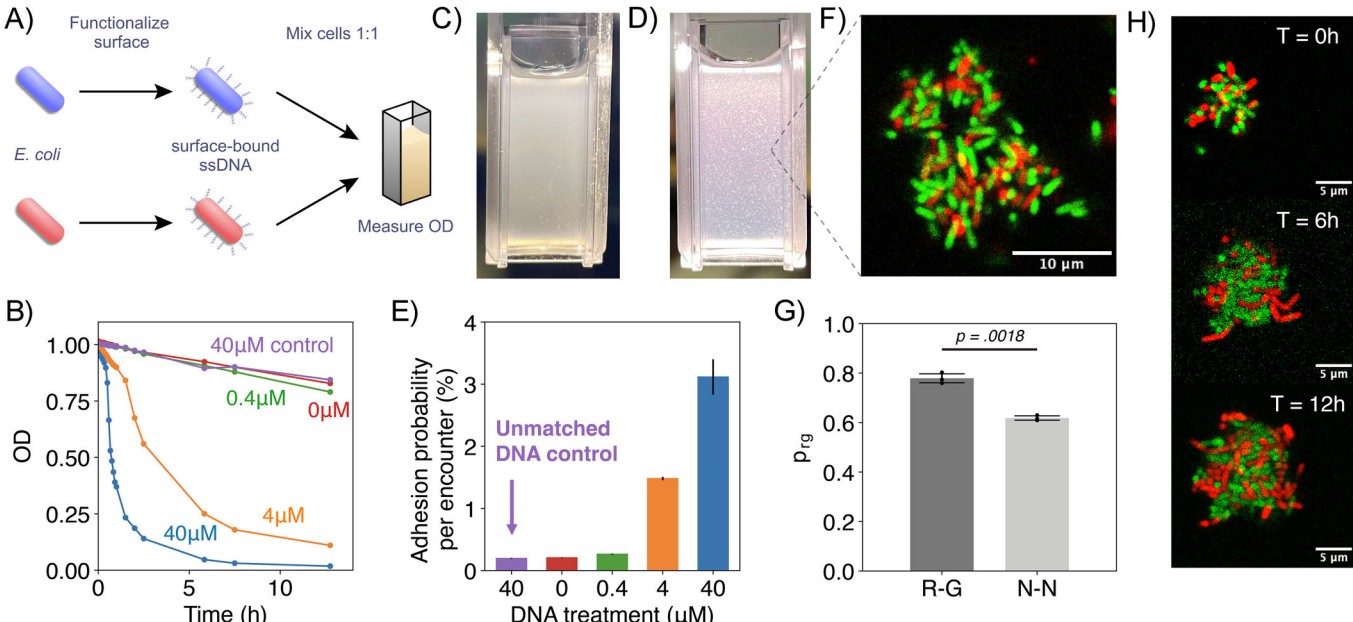

**Figure 5. Functionalizing cells with ssDNA to program cell-to-cell adhesion.**

(A) Schematic of sedimentation assay to quantify adhesive properties of cells. Two populations of cells are functionalized with ssDNA molecules and mixed 1:1 in a homogeneous solution whose $OD_{600}$ is measured over time. As the cells find each other in solution, complementary DNA strand pairs hybridize and induce the formation of cell aggregates that decrease the solution's $OD_{600}$. (B) $OD_{600}$ curves representing aggregation dynamics for cell populations modified with 0–40 μM of complementary DNA strands and a control condition modified with 40 μM of non-complementary DNA strands. Increasing treatment concentration of complementary DNA strands speeds up cell aggregation, causing faster $OD_{600}$ decay, while treatment with non-complementary DNA strands has no effect. (C, D) Pictures of cell populations treated with 0 μM of DNA (C) and 40 μM of complementary DNA strands (D) 1 h after mixing. Macroscopic aggregates are clearly visible in the treatment (D) while the solution remains turbid and homogeneous in the control (C). (E) Value of adhesion probability per encounter calculated from $OD_{600}$ curves in (B) according to a simplified aggregation model. Error bars are confidence intervals derived from the fitting procedure. (F–H) Microscopic characterization of DNA-guided aggregation. (F) Confocal slice showing internal structure of an aggregate formed by cells expressing green and red fluorescence and functionalized with complementary ssDNA strands. (G) Quantification of the fraction of cells inside aggregates with nearest neighbors of opposite color ($p_{rg}$). Two experimental conditions are measured: (1) (R-G) Cells expressing red fluorescence only adhere to cells expressing green fluorescence, like in (A); (2) (N-N) Cells expressing both fluorescence colors adhere to both types of cells. In each condition, three biological replicates were performed, and 10–20 aggregates were imaged in each replicate. Error bars represent SEM across biological replicates. Statistical significance was assessed with the two-tailed Welch's unequal variance t test. (H) Time-lapse of an aggregate embedded in LB-Agar culture medium showing growth over a period of 12 h. Source data are available online for this figure.

these results show that surface functionalization with ssDNA enables precise control over cell adhesion rate and strength based on the number of DNA molecules per cell. Further, this adhesion is programmable based on cell modification with the proper complementary DNA sequence.

We next asked whether this cell-to-cell adhesion method could be used to control the spatial structure of the formed cell aggregates at the single-cell level. To evaluate the micro-scale structure induced by our DNA hybridization-driven aggregation, we repeated our aggregation experiments with *E. coli* constitutively expressing red or green fluorescent proteins to enable them to be easily differentiated from one another as the two cell populations. After functionalizing each type of fluorescent cell with complementary DNA strands and mixing them for one hour (long enough for macroscopic aggregates to form), we imaged our samples with fluorescence confocal microscopy to determine the assembly structure and connectivity of the clusters generated.

Images of cell aggregates showed clusters ranging in size from 10 to 100 μm with a well-distributed mixture of green and red cells (Fig. 5F). Furthermore, a quantitative analysis of nearest-neighbor cell pairs showed that 78% ± 2% consisted of opposite-colored cells, a significantly higher mixing level than the 50% expected for a

random distribution ($P = 0.0021$, one-sample t test). In contrast, a control experiment in which both green and red fluorescent cells were functionalized with both DNA strands resulted in a significantly lower mixing fraction of 62% ± 1% ($P = 0.0018$, Welch's t test, Fig. 5G), confirming that the increased mixing was specifically driven by DNA-guided interactions. Furthermore, to ensure that aggregation did not compromise cell viability, we embedded aggregates in a nutrient-rich agar matrix and monitored their growth over time. Using this setup, we successfully recorded the growth of individual aggregates over time and saw division of cells uniformly across the cluster (Fig. 5H). These results demonstrate that ssDNA surface functionalization can be used to control the microscopic structure of microbial aggregates while maintaining the ability of cells to grow and divide.

## Discussion

We have developed a universal and modular chemical functionalization strategy for microbial cell surfaces that overcomes key limitations of existing approaches (Table 1). By leveraging a hetero-bifunctional crosslinker with a sulfo–NHS ester and a DBCO

moiety (Fig. 1), we achieved efficient, covalent attachment of azide-containing molecules to a broad range of live bacterial species, including Gram-negative, Gram-positive, aerobic and anaerobic species, without the need for genetic modification of cells or metabolic incorporation of labeling moieties (Figs. 2 and 3). Importantly, we demonstrate that efficient surface functionalization can be achieved while maintaining high cell viability, which is critical for any downstream applications (Fig. 3B). Furthermore, this approach allows quantitative control over the density of surface modifications, enabling the fine-tuning of phenotype strength (Figs. 4 and 5). We demonstrate this control for three distinct applications: fluorescence labeling for imaging (Fig. 2), beta-lactamase functionalization for antibiotic resistance (Fig. 4), and ssDNA-modification for controlled aggregate assembly (Fig. 5).

The ability to readily adapt our approach to different molecules underscores its modularity and flexibility. Moreover, the number of molecules tethered to the cell surface for different microbial strains was reproducible (Fig. 3C). This suggests that one could bypass extensive screening of new species by simply testing the few crosslinker concentrations corresponding to the required surface loading on the cell for a given application. This would largely simplify the extension of this method to untested species, overcoming the hardest obstacle to the method's widespread adoption.

The potential of this method to attach enzymes to cell surfaces and thereby extend native biological functions is of particular interest for microbiome-based therapies and bioremediation. Regarding microbiome therapies, the use of oral antibiotics can result in the loss of gut diversity, increasing susceptibility to diseases such as *C. difficile* infection (Deshpande et al, 2013). To avoid this, the prescription of probiotics during antibiotic treatment is a common practice (Goldenberg et al, 2017). However, ingested probiotics are also sensitive to antibiotic treatments, preventing efficient reconstruction of the intestinal microbiota (Pan et al, 2022). Using our method, it is possible to confer antibiotic resistance to microbes in a non-genetic manner (Fig. 4B), enabling the preparation of antibiotic-resistant probiotic strains that can be taken as supplements alongside treatments to maintain gut diversity. On the side of bioremediation, spatially structured and surface-functionalized microbial communities have been shown to be more effective at removing pollutants from the environment (Pan et al, 2019; Kim et al, 2011). In these applications, the significantly higher surface functionalization efficiency of the method described here can be directly translated into longer stability of engineered functions, reducing costs associated with the manufacturing of new batches of functionalized cells.

Our surface modification approach further provides a means to precisely control bacterial adhesion (Fig. 5), an essential phenotype in microbial community assembly that can lead to complex ecological interactions (Schluter et al, 2015). By functionalizing cells with increasing densities of ssDNA, we achieved tunable adhesion strengths, with adhesion probabilities per cell encounter increasing up to 15-fold for cells modified with the highest densities of ssDNA (Fig. 5E). The sequence specificity of this approach was also validated using non-complementary DNA, which showed adhesion levels comparable to untreated samples, confirming that DNA hybridization is responsible for the assembly. Fluorescence microscopy further revealed that ssDNA-mediated aggregation

promotes a highly mixed spatial distribution of bacterial populations (Fig. 5F), with mixing levels significantly higher than a random assortment (Fig. 5G), and that cells remained viable post-aggregation (Fig. 5H), ensuring that this strategy does not compromise microbial growth. These findings highlight the potential of ssDNA functionalization as a tool for engineering structured microbial consortia, which could be leveraged as seeding communities for microbiome modulation, biofilm engineering, and biotechnological processes requiring precise cellular assembly.

Despite these advantages, some limitations remain. While our approach allows functionalization of a wide range of microbes, it still requires some empirical optimization for each new species, as demonstrated by the varying sensitivity of different bacterial strains to the crosslinker. In addition, since our method does not rely on genetic modifications, it is not stable in the long term, and functionalized cells will eventually lose their endowed functions. As the number of surface molecules is halved at each cell division (Fig. EV2), the major factors determining how long cells can maintain their phenotypes are the cell's growth rate, the minimum number of surface molecules needed to display a phenotype, and the phenotype strength required in a given application. When using functionalized cells, it is important to explore all three of these factors to determine whether cells will be functional long enough to have the desired impact.

Overall, this work establishes a simple yet powerful framework for microbial surface engineering that is broadly applicable across biological and biotechnological contexts. By decoupling surface functionalization from genetic and metabolic constraints, this approach paves the way for the development of next-generation live therapeutics and natural microbial consortia with extended biological functions. As surface functionalization continues to evolve, integrating this method with other biorthogonal chemistries and adaptive biomaterials could further expand its potential, opening new frontiers in synthetic biology and microbiome engineering.

# Methods

**Reagents and tools table**

| Reagent/resource | Reference or source | Identifier or catalog number |
| --- | --- | --- |
| **Experimental models** | | |
| *Escherichia coli* MG1655 | ATCC | 700926 |
| *Vibrio splendidus* | Datta et al, 2016 | 1A01 |
| *Neptunomonas phycotrophica* | Datta et al, 2016 | 3B05 |
| *Bacteroides ovatus* | ATCC | 8483 |
| *Lactococcus lactis* | Vatanen et al, 2016 | RJX1236 |
| *Lactobacillus rhamnosus* | Vatanen et al, 2016 | RJX1228 |
| **Recombinant DNA** | | |
| pEB2-mScarlet-I | Addgene | 104007 |
| pEB2-mTurquoise | Addgene | 103972 |

| Reagent/resource | Reference or source | Identifier or catalog number |
|---|---|---|
| **Oligonucleotides and other sequence-based reagents** | | |
| /5AzideN/ CACACACACACACACACACA | IDT | /5AzideN/ |
| /5AzideN/ TGTGTGTGTGTGTGTGTGTG | IDT | /5AzideN/ |
| **Chemicals, enzymes, and other reagents** | | |
| Miller's Luria Broth | Sigma-Aldrich | L3522 |
| S-Gal®/LB Agar Blend | Sigma-Aldrich | C4478 |
| Marine Broth 2216 | Sigma-Aldrich | 76448 |
| Brain Heart Infusion Broth | Sigma-Aldrich | 53286 |
| L-Cysteine | Sigma-Aldrich | C7352 |
| 1:100 Hemin & Vitamin K solution | BD biosciences | 212354 |
| Sodium phosphate dibasic | Sigma-Aldrich | S9763 |
| Potassium phosphate monobasic | Sigma-Aldrich | P0662 |
| Sodium chloride | Sigma-Aldrich | S9888 |
| Ammonium chloride | Sigma-Aldrich | 213330 |
| Magnesium sulfate heptahydrate | Sigma-Aldrich | 1.05886 |
| Calcium Chloride dihydrate | Sigma-Aldrich | C3306 |
| Glucose | Sigma-Aldrich | G8270 |
| TWEEN® 20 | Sigma-Aldrich | P1379 |
| DBCO–sulfo–NHS | Sigma-Aldrich | 762040 |
| NHS-azide | Thermo Fischer | 88902 |
| Alexa Fluor 488 azide | Thermo Fischer | A10266 |
| Penicillinase from Bacillus cereus | Sigma-Aldrich | P0389 |
| SYTOX™ Orange | Thermo Fischer | S11368 |
| Quantum™ Alexa Fluor® 488 MESF kits | Bangs Laboratories | 488B |
| Ampicillin Sodium Salt | Sigma-Aldrich | A0166 |
| **Software** | | |
| Ilastik | Berg et al, 2019 | |
| **Other** | | |
| BD FACSMelody™ Cell Sorter | BD Biosciences | |
| Tecan Spark® Multimode Microplate Reader | Tecan | |
| BioTek Synergy H1 Multimode Reader | Agilent | |
| Agilent BioTek Epoch microplate spectrophotometer | Agilent | |
| 7K MWCO Zeba spin desalting columns | Thermo Fisher | 89877 |
| ImageXpress Micro Confocal | Molecular Devices | |
| Zeiss LSM 710 Laser Scanning Confocal | Zeiss | |

## Methods and protocols

### Culture media

Experiments were performed with the following media: Miller's Luria Broth (LB, Sigma-Aldrich). Marine Broth 2216 (MB, Sigma-Aldrich). Brain Heart Infusion Broth (BHI, Sigma-Aldrich) supplemented with 0.05% L-Cysteine (Sigma C7352) and 1:100 Hemin & Vitamin K solution (BD 212354), referred to as BHIS. M9 medium composed of M9 salts (47.76 mM $Na_2HPO_4$, 22.04 mM $KH_2PO_4$, 8.56 mM NaCl, and 18.69 mM $NH_4Cl$ premixed) supplemented with 2 mM $MgSO_4$, 0.1 mM $CaCl_2$, 45 mM Glucose, and 0.01% Tween-20 (all from Sigma-Aldrich). A chemically defined minimal medium for anaerobic growth, termed ZM media, was prepared as described previously (Henke et al, 2019; Moreno-Gámez et al, 2025).

### Strains and growth conditions

*Escherichia coli* MG1655 (grown aerobically at 37 °C in LB or M9 medium), *Vibrio splendidus* 1A01 (grown aerobically at 25 °C in Marine Broth 2216), *Neptunomonas phycotrophica* 3B05 (grown aerobically at 25 °C in Marine Broth 2216), *Bacteroides ovatus* ATCC8483 (grown anaerobically at 37 °C in BHIS and ZM medium), *Lactococcus lactis* RJX1236 (grown anaerobically at 37 °C in BHIS and ZM medium), *and Lactobacillus rhamnosus* RJX1228 (grown anaerobically at 37 °C in BHIS and ZM medium). Plasmids pEB2-mScarlet-I and pEB2-mTurquoise were acquired from Addgene and transformed into *E. coli* MG1655 to create constructs constitutively expressing red and green fluorescent proteins, respectively.

### Surface functionalization protocol

1) Cells from a frozen stock were streaked onto 1.5% Agar plates made with compatible growth media (LB, MB, or BHIS).
2) Once colonies started to show, an isolated colony was picked and cultured in compatible liquid media (M9, MB, or ZM) overnight.
3) Cells were then washed twice in phosphate buffer saline (PBS, pH 7.4) via centrifugation at $5000 \times g$ for 1 min to remove any free amines present in the liquid medium. Once washed, cells were concentrated to an optical density between 1 and 10 in a volume of 1 ml.
4) The crosslinker solution was then prepared by weighing a small amount of Dibenzocyclooctyne-sulfo-N-hydroxysuccinimidyl ester (DBCO–sulfo–NHS from Sigma-Aldrich, 762040) and solubilizing it in PBS at a concentration of 1000 mg/L at room temperature right before use. The crosslinker was then added to the cell solution to a final concentration varying from 0.064 to 1000 mg/L, depending on the experiment.
5) This mixture was incubated at 25 °C for 1 h.
6) Once the incubation was complete, cells were washed twice with PBS via centrifugation to remove excess crosslinker and resuspended in 100 µl of PBS.
7) Then, a 1 mM solution of azide-modified molecules was prepared. In our experiment, this solution consisted of either Alexa Fluor 488 azide (A10266, Thermo Fischer) prepared in DMSO, azide-modified penicillinase from *B. cereus* (P0389, Sigma-Aldrich, modified as described in Methods) prepared in

mili-Q water, or 5' azide-modified oligos (modification code/ 5AzideN/, IDT) also prepared in mili-Q water.

8) Following preparation of the azide-modified molecule solution, 4 µl of this solution was added to the 100 µl of cell suspension, making the final concentration of azide-modified reagent 40 µM. Lower concentrations were used in some cases to tune the number of molecules loaded to the cell surface, as described in the text.

9) This mixture was then incubated at 25 °C for 1 h.

10) After that, cells were thoroughly washed with 1 ml of PBS at least four times to remove unreacted molecules from the solution.

11) The final cell suspension was used in downstream applications.

### Single-cell fluorescence quantification with flow cytometry

Cells were analyzed with a BD FACSMelody™ Cell Sorter equipped with two lasers (488 nm, 561 nm). Before analysis, treated cells were diluted in PBS buffer to an appropriate density and when necessary incubated with 0.1 µM of SYTOX™ Orange Nucleic Acid Stain (Thermo Fisher) for 15 min. Gates were set based on only buffer controls to exclude non-cell events, and doublet discrimination was performed with FSC-H and FSC-A channels. Both the Alexa Fluor 488 and the SYTOX™ Orange dyes could be simultaneously excited and have their emissions quantified with minimal crosstalk between the two fluorescent channels. At least 10,000 meaningful events were recorded for each analyzed sample.

### Flow cytometry fluorescence channel calibration

Calibration curves for the conversion between fluorescence intensity and number of molecules were constructed with Quantum™ Alexa Fluor® 488 MESF kits from Polysciences according to manufacturer instructions. The kit contained five types of beads with different numbers of Alexa Fluor 488 molecules attached to them. The measurements were performed by diluting the fluorescent beads from the kit in PBS and quantifying green fluorescence in the flow cytometer. An appropriate PMT voltage for the fluorescent channel was set to ensure that all beads fell within the linear range of the equipment (Appendix Fig. S2a). Those PMT voltages were maintained for all subsequent sample measurements, allowing a quantitative comparison of fluorescence values.

The calibration curve was constructed by calculating the difference between the measured fluorescence of each bead sample and the measured fluorescence of non-modified beads to account for any background fluorescence of the beads themselves. These values were then regressed against the number of molecules per bead reported by the kit (Appendix Fig. S2b). The resulting regression coefficients were used as conversion factors between normalized fluorescence intensity values and Molecules of Equivalent Soluble Fluorochrome (MESF) units for our measurements of cell fluorescence. When applying this calibration curve to the cells, measurements were also normalized by subtracting the background fluorescence of unmodified cells exposed to the fluorophore to make sure our estimation of the number of surface-attached fluorophores did not include any background fluorescence from the cells, nor any fluorophores attaching to the cells non-specifically. If azide-modified molecules were labeled with more than one fluorophore, our estimate of the number of surface-attached fluorophores was divided by the degree of labeling of the molecule

to arrive at the number of surface-attached molecules that is reported in the manuscript.

Overcrowding of fluorophores on a surface tends to decrease the overall fluorescence in a non-linear fashion. Our calibration with beads partially accounts for this phenomenon, but could possibly underestimate the number of molecules on the surface of the cell, as the beads had a radius of 7–9 µm while cells are usually 1–2 µm long. Therefore, our estimates of the number of molecules on the cell surface should be taken as a lower bound.

### Viability estimation from nucleic acid staining

Samples stained with SYTOX™ Orange showed a clear separation into a group with high fluorescence and a group with low fluorescence. Therefore, a simple threshold value was used to classify recorded events into viable (low fluorescence) or non-viable (high fluorescence). The percentage of viable events was then taken as a proxy for the percentage of viable cells in the sample, and the sample viability was estimated as the ratio of its percentage of viable cells to that of the control condition.

### Activity estimation from regrowth dynamics

To estimate bacterial activity, overnight cultures were grown in their respective minimal medium, except for the two marine strains (1A01 and 3B05), which were grown in Marine Broth, a rich medium. The next day, cultures were washed twice with 1× PBS via centrifugation. Then, 50 µl of the washed cell solution was added to a 96-well plate and mixed with 50 µl of a 2× DBCO–sulfo–NHS solution, resulting in the final DBCO–sulfo–NHS concentrations shown in Fig. 3. Each of the well solutions was then mixed by pipette mixing and left incubating for 1 h at room temperature. After incubation, the well solutions were mixed again, and two 40× dilutions were performed by taking 5 µl of the well solution into 195 µl of PBS and then 5 µl of this new solution into 195 µl of growth media. The 96-well plate containing the samples diluted in growth media was then taken to our Tecan plate reader, and OD was recorded at a 600 nm wavelength.

After recording the regrowth dynamics of all strains under different conditions, we performed two fitting procedures to estimate the initial $OD_{600}$ of these samples. For strains grown in minimal media, we fit an exponential growth model by computing a linear regression of the log of $OD_{600}$ measurements versus time during the exponential phase of growth. We then estimated the exponential model parameters, growth rate, and log of initial $OD_{600}$ (OD0), as the slope and intercept parameters of the fit, respectively.

For strains grown in rich media, we fit a logistic growth model by computing a linear regression of the percentage $OD_{600}$ change (OD/OD) versus the $OD_{600}$ measurements. We then estimated the logistic model parameters, growth rate, and carrying capacity, as the intercept of the fit and the negative of the ratio of slope to intercept of the fit, respectively. Inputting these parameter estimates and the initial data point used for fitting into the analytical solution of the logistic equation, we finally calculated the initial $OD_{600}$ of the sample:

$$OD_0 = k \cdot OD(t_0)/(OD(t_0) + (k - OD(t_0))e^{rt_0})$$

For both of these fitting procedures, we verified that the $r^2$ of the linear regressions was above 0.99 for all samples and all conditions.

After estimating the initial $OD_{600}$ for each sample from these fitting procedures, viability was calculated as the ratio of the

sample's initial $OD_{600}$ to that of its paired, untreated sample. This ratio was calculated for three to four independent biological replicates for each strain as described in the main text.

### Estimation of surface modification dilution with growth

*E. coli* cells were functionalized with Alexa Fluor 488 azide using the optimal functionalization conditions determined here. After modification, cells were mixed with a 0.5% LB Agar solution at 40 °C and pipetted into a glass-bottom 96-well plate. Cells were then imaged with a fluorescence confocal microscope over time and z-stacks were recorded at each time point. To analyze the images, cell lineages were cropped individually and for each of them, the brightest z-slice and the two adjacent z-slices were averaged before cells were segmented using a global threshold. Segmented regions were then used to calculate the average fluorescence and area values used to fit the exponential decay function. The number of area doublings was calculated as the logarithm base 2 of the ratio of the lineage area at a certain time point t to the initial lineage area.

$$d = log_2(Area(t)/Area(t_0))$$

Fitting was performed with the curve_fit function from the scipy library. This procedure was performed for six biological replicates.

### Azide modification of Penicillinase from B. cereus

1) Penicillinase from *B. cereus* was purchased from Milipore Sigma (P0389) and kept at 4 °C as a powder until use.
2) When needed, a stock solution was prepared by diluting the powder in PBS at a concentration of 10 mg/ml or approximately 0.4 mM.
3) In addition, a solution of NHS-azide (Thermo Fischer, 88902) was prepared in DMSO at a concentration of 4 mM.
4) To begin the azide modification reaction, 100 μl of the stock protein solution was mixed with 25 μl of the NHS-azide solution and incubated at room temperature for 30 min.
5) At the end of the incubation, 10 μl of 0.5 mM TRIS buffer was added to quench the reaction.
6) The final mixture was then purified using 7 K MWCO Zeba spin desalting columns (Thermo Fisher, 89877).
7) Final products were analyzed with a Maldi-TOF showing an increase of 1800 m/z in mass.
8) The purified, azide-tagged proteins were then kept at 4 °C for at most a week until use.

### Minimum inhibitory concentration assay

To measure minimal inhibitory concentration (MIC), 100 μl of a serial dilution of ampicillin was prepared in 96-well plates with concentrations ranging from 3200 μg/ml to 3.125 μg/ml in 2× dilution steps and a last condition with 0 μg/ml of ampicillin, totaling 12 experimental conditions. To each of these solutions, 100 μl of a cell solution was added, making the final ampicillin concentration range from 0 μg/ml to 1600 μg/ml in the experiments. These 96-well plates were then incubated at 25 °C under shaking conditions, and the final $OD_{600}$ was measured after 24 h.

For the experiments matching surface-bound to free-floating enzyme concentrations, the total concentration of enzymes was estimated by assuming that functionalized cells have an average of 210,000 enzyme molecules on their surface, in agreement with our fluorophore estimations, and that 1 ml of an $OD_{600}$ 1 *E. coli* cell solution contains $10^9$ cells. Therefore, the concentration of enzymes in an $OD_{600}$ 0.1 solution of functionalized cells was estimated to be 35 nM. To modulate the concentration of enzymes in solutions of functionalized cells, we inoculated them at $OD_{600}$ of 0.1, 0.01, 0.001, and 0.0001. Matching these conditions, solutions of non-functionalized cells with the same $OD_{600}$ were prepared in media containing 35, 3.5, 0.35, and 0.035 nM of the same azide-modified beta-lactamases used for cell functionalization.

### Sedimentation assay

Starting with overnight cultures of *E. coli* cells, we functionalized them with the following 20-nucleotide-long, azide-modified ssDNA molecules (ordered from IDT):

/5AzideN/CACACACACACACACACACA
/5AzideN/TGTGTGTGTGTGTGTGTGTG.

For surface functionalization, we used the protocol previously described, with the only exception that the concentration of the ssDNA solution used during incubation was dependent on the experimental condition. We prepared incubations with 0, 0.4, 4, and 40 μM ssDNA solutions for each of the two DNA sequences. After completing the surface functionalization protocol, the $OD_{600}$ of each sample was standardized to 1. Then, solutions of cells functionalized with complementary DNA strands at the same concentration were mixed to prepare the experimental conditions referred to as 0, 0.4, 4, and 40 μM in the main text. In addition, two samples of cells independently functionalized with 40 μM solutions of the same DNA sequence were mixed to prepare the experimental condition referred to as 40 μM control in the main text. The $OD_{600}$ of these five mixed solutions was then measured with a spectro-photometer, initially every 5 min and later at longer intervals of time.

### Estimation of adhesion probability per encounter

The data collected from the sedimentation assay shows a decrease in $OD_{600}$ due to cellular aggregation. Assuming that the $OD_{600}$ of the solution is proportional to the number of cell aggregates present, we can describe this $OD_{600}$ decrease by modeling the aggregation dynamics of the cells. To develop this aggregation model, we make a further simplifying assumption and only take into account first-order reactions, in which single cells bind together to form two-cell aggregates. This is a reasonable assumption at the beginning of the assay when most cells are disaggregated but it breaks down later when higher-order reactions lead to the formation of multicell aggregates.

In this simplified aggregation model, reactions happen between green (G) and red (R) cells at a rate k, and so the concentration of red and green cells in solution follows the differential equation:

$$d[G]/dt = d[R]/dt = -k[R][G]$$

Assuming that both cells start at the same concentration $C_0$, we can solve these equations to find the concentration of each cell type as a function of time:

$$[G(t)] = [R(t)] = C_0/(C_0kt + 1)$$

And since all cells that react become two-cell aggregates, the concentration of the latter follows:

$$[RG(t)] = C_0 - [R(t)]$$

So, the overall $OD_{600}$ of the solution at a time $t$ can be calculated to be proportional to:

$$OD(t) \propto [G(t)] + [R(t)] + [RG(t)] = C_0 + C_0/(C_0 kt + 1)$$

Where the initial $OD_{600}$ is proportional to $2C_0$, the total initial concentration of cells. Measuring cell concentration in $OD_{600}$ units, the proportionality constant in the previous equation becomes 1, and we can fit the derived equation to the $OD_{600}$ lines measured with the sedimentation assay directly. Performing the fit for the data points within the first 30 min to 1 h of sedimentation, we were able to estimate the reaction rate $k$ between the cells. Assuming the reactions are driven by diffusion, we can rewrite the reaction rate as $k = 4\pi D_{eff} R_c p$, where $D_{eff} = 2D$ is the effective diffusion constant between two microbial cells calculated as the sum of the diffusion constants of each cell, $R_c = 2R$ is the encounter radius calculated as the sum of the radii of each of the cells, and $p$ is the probability cells adhere to each other after an encounter. Finally, using the Stokes-Einstein relation ($D = k_B T/6\pi R\eta$) for the diffusion constant of a single microbial cell, we arrive at the following formula relating the reaction rate $k$ to the adhesion probability per encounter $p$:

$$k = 16\pi DR \cdot p = 8k_B T/3\eta \cdot p$$

Using this equation and our estimates for the reaction rate of the cells, we were able to estimate the adhesion probability per encounter in our system.

### Quantification of microscopic structure in cell aggregates

To quantify the microscopic structure of cell aggregates, aggregated cells were resuspended in an agar matrix and imaged with a confocal microscope. Individual confocal slices were then segmented with Ilastik (Berg et al, 2019), and further image analysis was done in Python. Cell positions were calculated as the centroid of the segmented objects in the images, and pairwise Euclidean distances were used to determine nearest-neighbor pairs. Then, the fraction of opposite colored pairs was calculated.

## Data availability

This study includes no data deposited in external repositories.

The source data of this paper are collected in the following database record: biostudies:S-SCDT-10_1038-S44320-026-00202-z.

## Peer review information

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

## Acknowledgements

We thank all members of the Cordero Laboratory and Simons PriME collaboration for continued discussions and input. We acknowledge funding from the Simons Foundation through the collaboration on Principles of Microbial Ecosystems (PriME) award number 542395 (OXC). GTV is grateful for support from the Hugh Hampton Young Memorial Fund Fellowship.

## Author contributions

**Gabriel T Vercelli**: Conceptualization; Investigation; Methodology; Writing—original draft; Writing—review and editing. **Xingcheng Zhou**: Methodology. **Stefany Moreno-Gámez**: Investigation. **Rashi R Jeeda**: Methodology. **Rachel Gregor**: Investigation. **Jonasz Słomka**: Investigation; Methodology. **Akorfa Dagadu**: Investigation. **Ariel L Furst**: Conceptualization; Supervision. **Otto X Cordero**: Conceptualization; Supervision; Writing—original draft; Project administration; Writing—review and editing.

Source data underlying figure panels in this paper may have individual authorship assigned. Where available, figure panel/source data authorship is listed in the following database record: biostudies:S-SCDT-10_1038-S44320-026-00202-z.

## Disclosure and competing interests statement

GTV, OXC, and ALF are inventors on U.S. provisional patent application No. 63/836,388, filed by the Massachusetts Institute of Technology and covering the full scope of the work presented here.

# Expanded View Figures

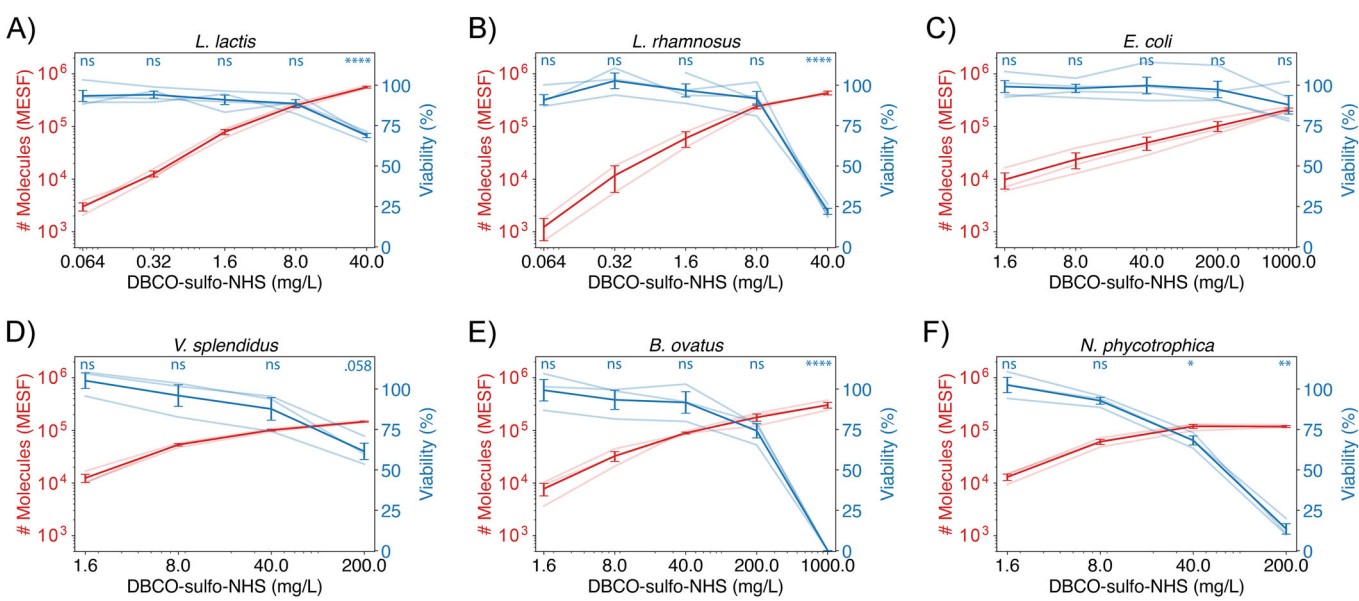

**Figure EV1. Quantification of efficiency and viability trade-off for the six strains screened.**

(A-F) Semi-transparent curves represent different biological replicates. Significance of viability effects are displayed on top of graphs and were calculated with a one-sided, one-sample $t$ test with $n = 3$ or 4 depending on the number of biological replicates in the sample. $P$ values were corrected using the Benjamini–Hochberg procedure. The meaning of the symbols is $P > 0.1$ (ns), $0.1 > P > 0.05$ (displayed), $0.05 > P > 0.01$ (*), $0.01 > P > 0.001$ (**), $0.001 > P > 0.0001$ (***), and $0.0001 > P$ (****). Source data are available online for this figure.

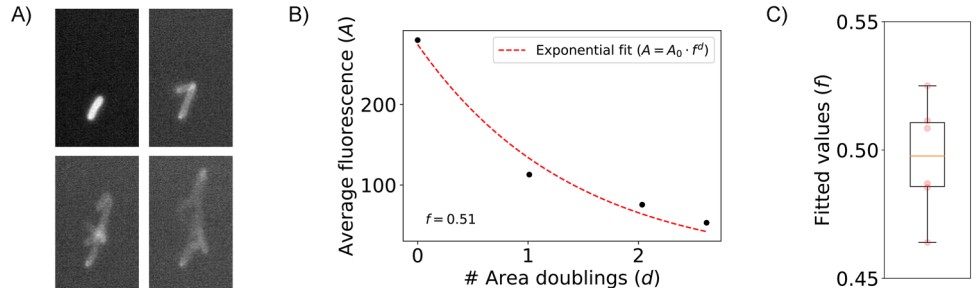

**Figure EV2.   Quantification of the dilution of surface modification with growth.**

(A) Microscopy images showing the growth of a lineage of cells functionalized with fluorophores. (B) Average cell fluorescence (A) as a function of area doublings ($d = \log_2(\mathrm{area}/\mathrm{initialarea})$). Cells were segmented from images in (A) to calculate average fluorescence and area values. An exponential fit was then performed to estimate the fraction of fluorescence retained after each area doubling (f). (C) Fraction of fluorescence retained after every doubling (f). A box plot represents the first and third quartiles of six biological replicates with a line at the median and whiskers representing maximum and minimum values. Source data are available online for this figure.

