## [Peer Review File · Molecular Systems Biology]

A universal surface functionalization technique to chemically enhance live microbial cells

Gabriel Vercelli, Xingcheng Zhou, Stefany Moreno-Gómez, Rashi Jeeda, Rachel Gregor, Jonasz Słomka, Akorfa Dagadu, Ariel Furst, and Otto Cordero

Corresponding author(s): Otto Cordero (ottox@mit.edu), Ariel Furst (afurst@mit.edu)

Review Timeline:

Transfer Date:	23rd Oct 25
Editorial Decision:	11th Dec 25
Revision Received:	21st Jan 26
Editorial Decision:	29th Jan 26
Revision Received:	31st Jan 26
Accepted:	11th Feb 26

Editor: Poonam Bheda

Transaction Report:

The first round of review of this manuscript was performed at another journal.

Reviewer #1 (Remarks to the Author):

I appreciate the detailed revisions and agree with most of it. However, I still do not agree that it is out of scope to provide estimates for the duration of the labeling simply because everyone would use a different phenotype. Since the goal is to reach a wide audience with exactly that purpose, the authors can still benchmark this.

We have added a new supplementary figure (Sup. Fig. 8) showing the decay of fluorescent signal as a function of generation time. As expected, signal intensity decreases by approximately half after each division.

In the specific case of fluorophores, and with a microscope setup not optimized for this application, the fluorescent signal becomes undetectable after three generations. In general, the number of generations required for the phenotype to disappear will depend on the application (e.g., certain enzymes may confer a detectable phenotype with only a few surface-attached copies).

Overall, while I still remain excited about the technology, I am not convinced that it really would be easily adaptable to anyone in the field without the chemistry or chemical engineering knowledge of authors. Also, the the fundamental advances consistent with this journal seem somewhat misaligned with the nature of the manuscript.

Based on our experience training students with no chemical engineering background, we are confident that the technique is easily adaptable. The entire process takes only a few hours and consists of a simple series of incubations and washes.

Reviewer #2 (Remarks to the Author):

I have carefully read the authors' response and the revised version of the manuscript, which has indeed improved. Nevertheless, I still believe that although this is a solid piece of work that will contribute to the area of microbe modifications, it does not demonstrate sufficient novelty to merit publication in [journal name redacted]. The main reason is that the chemistry employed is well known and has already been applied for cell surface modifications. For example, a similar two-step protocol for modifying cells was reported in New Water-soluble Alkynylating Agent for Cell Surface Protein: Sulfosuccinimidyl 4-Pentynoate. Bull. Korean Chem. Soc. 2013, 34(6), 1895–1898. I believe that the comparison with previous work will be of value to scientists in this field. At the same time, it should be recognized that systematic optimization through variations in structure or experimental conditions is a common practice. Previous studies did not emphasize this aspect, largely because such fine-tuning was not considered essential once the core methodology had proven effective. Thus, while achieving a 4–50-fold increase in the number of molecules per bacterium is notable, such improvements arising from optimization are not particularly surprising.

It is correct that there are numerous permutations of click-chemistry techniques that employ components of our protocol. However, the key point remains that our technique

is the first designed specifically for application to live bacterial cells while maintaining maximal viability—an essential requirement for downstream applications. The paper referred to by the reviewer uses a copper-catalyzed click-chemistry technique that is well known to be highly toxic to cells. Its goal is to perform protein surface labeling and not live-cell modification. This is a great example of how subtle variations in structure and experimental conditions can have devastating consequences for the effects of the technique on cell viability. By systematically optimizing these conditions we not only achieve a 50-fold enhancement on labeling efficiency, we are able to endow cells with phenotypes that are impossible to achieve without that high degree of labeling such as the macroscopic cell aggregation and non-genetic antibiotic resistance described in the paper.

Finally, the authors' claim that “the reason why the applications remain limited is precisely because current methods have low efficiency, or impose drastic effects on cell viability, or are hard to translate to other organisms, all of which are the issues that we carefully addressed in this work” is, in my view, far too strong.

In our experience, the main challenge lies in quantifying the efficiency of the modifications and assessing their impact on cell viability, among other effects. This is a defining aspect of our work. Thus, the issue is not merely one of optimization, but also of developing quantitative assays that allow us to understand the technique and its effects in detail. To our knowledge, no previous study has implemented quantitative assays to the extent presented in our paper.

That said, this is a carefully executed and well-presented study. I believe it will be of genuine interest to researchers working on bacterial engineering and may find a more appropriate home in a specialized journals, where it can receive the attention it deserves.

Reviewer #3 (Remarks to the Author):

This manuscript presents a surface functionalization method using DBCO-sulfo-NHS crosslinkers to attach azide-containing molecules to bacterial surfaces, and provides quantitative, non-genetic control over microbial phenotypes and complements genetic engineering approaches. However, the use of NHS ester-mediated amine coupling combined with SPAAC chemistry for microbial surface modification has been already well-established. The only difference in this work is that DBCO is attached to the bacterial surface instead of N_3 , with N_3 incorporated into the functional module. This does not constitute a genuine methodological innovation in surface modification. Therefore, despite of improvement of this method, the reviewer still believes that from the perspective of innovation, depth of investigation, and completeness of the data, this work does not meet this journal's standards. More major concerns are as below.

As acknowledged earlier, there are multiple permutations similar to our technique. However, our claim is not to have discovered a new chemistry, but rather to have established a protocol that maximizes cell viability in live bacteria—including Gram-

negative, Gram-positive, marine, and gut bacteria—under both aerobic and anaerobic conditions. We have also developed new quantitative assays that allow us to benchmark our technique and its consequences for cell viability with a level of precision absent in previous studies. Nonetheless, we understand that the question of novelty can be a matter of perception.

1. The authors claim that DBCO-sulfo-NHS enables efficient modification through amine-specific reactions but fail to provide critical kinetic data. Why were surface plasmon resonance (SPR) and isothermal titration calorimetry (ITC) not used to determine the reaction rate constants of DBCO-sulfo-NHS with free amines, and how do these compare with traditional NHS esters? What is the quantitative difference in reaction specificity (towards amines versus other functional groups)?

It is clear from this and the following questions that the reviewer expects this paper to be something it was not intended to be. Most of the requests are unreasonable, as they would entail an enormous amount of additional work without adding clarity or strengthening the main message of the paper.

Why were reaction constants not measured? Because this is not a protein biochemistry study of the DBCO-sulfo-NHS + amine reaction. Such measurements are not typically reported in the relevant literature and would require substantial additional effort without advancing the primary goal of our study—to develop a molecular engineering technique for live bacterial cells.

2. DBCO has inherently poor solubility; although the authors employ a sulfonated DBCO derivative to mitigate this, the NHS-sulfo-DBCO moiety is sterically bulky. It is therefore necessary to assess the upper limit of DBCO surface loading and its correlation with bacterial viability.

We disagree that this is necessary. Measuring the upper limit of DBCO surface loading would not improve our conclusions, as we have already established, quantitatively, an upper limit of reactant concentration to maintain bacterial viability and its corresponding surface loading. We also believe the reviewer is thinking of sulfo-NHS-DBCO as a sterically bulky molecule because they are thinking about internal cell labeling techniques. Indeed, DBCO molecules are too large to permeate through the membrane and would represent a limitation if the labeled molecule had to be incorporated into the cell, like what is done in standard metabolic labeling techniques that use either azides or alkynes (much smaller chemical moieties). However, here it is in our advantage that the labeling molecule doesn't get inside the cell since we are focused on surface labeling and, for that use, DBCO molecules are small and do not pose steric issues for labeling surface proteins.

3. A direct comparison is needed between NHS-mediated surface modification with DBCO versus N_3 in terms of labeling efficiency, cell viability, molecular distribution, and stability. Without such data, the novelty and necessity of the work remain unclear.

Again, the reviewer requests additional, complex experiments without clear benefit to our study. We have conducted a thorough quantitative comparison against previous work and have established the impact of our modifications on cell viability and labeling efficiency. Stability is addressed in the revised version by analyzing the rate of fluorescence signal decay as a function of generation number in cells loaded with fluorophores (Sup. Fig. 8).

4. All tested strains are of "easily modifiable" types (e.g., non-capsulated, low lipopolysaccharide). Key clinical/environmental strains, such as capsulated *Klebsiella pneumoniae* or mycobacteria, remain entirely untested.

This is a surprising comment. We use natural isolates of marine bacteria, including Gram-positive, Gram-negative, aerobic, and anaerobic strains. These are not "easily" modifiable, but rather representative of the bacteria commonly used in synthetic biology and molecular engineering. Naturally, some strains remain untested—and always will. Each bacterial species will exhibit specific efficiency/viability trade-offs, some low and some high, but it is unclear how testing additional organisms would advance the paper or where such testing should reasonably stop. If we were to test *Klebsiella*, would another reviewer request *Planctomycetes* or a methanogen?

5. The accuracy of molecular quantification methods lacks cross-validation. The authors estimated the number of molecules on the cell surface using fluorescent beads but acknowledge potential underestimation due to geometric differences. However, no independent methods were used to validate the quantification results, casting doubt on the reliability of critical data such as the number of molecules on bacteria. If the quantification is inaccurate, efficiency comparisons become meaningless.

It is standard practice in the field to use analogs and standards for quantification. Although this approach is not perfect, there is currently no standard method to quantify the number of molecules attached to a cell surface. When such measurements are needed, they must be calibrated for each specific application.

6. Quantitative data on functional stability are missing, raising doubts about practical application value. Although the authors acknowledge the short-term nature of non-genetic modifications, they have not experimentally demonstrated the duration of modification or the half-life of key phenotypes.

In the revised version, we show that the signal decays by approximately half with each generation. The total number of generations until the phenotype disappears is application-dependent. With our (suboptimal) microscope setup, the fluorescent signal per cell becomes undetectable after three divisions. However, many fluorophores remain attached to each cell, even if their signal is too weak to be detected by our instrument.

7. The authors observed that Gram-positive bacteria exhibit higher modification efficiency but greater sensitivity to the crosslinker (Fig. 3c, d), yet this remains merely a

phenomenological description. Critical questions remain unaddressed experimentally: Do Gram-positive and Gram-negative bacteria differ significantly in the content of surface-exposed free amines? Supplementary quantitative experiments are required.

We acknowledge that our description is phenomenological, and we are comfortable with that. The reviewer cannot reasonably expect every observation to be substantiated with mechanistic detail, as each would require an extensive study beyond the scope of this research.

8. More tests should be used to present the successful of modification. Such as NTA, zeta potential, SEM, and TEM.

This is another unreasonable request without a clear motivation.

9. More solid data should be added for the surface functionalization. For example, the tests of enzyme activity of beta-lactamase. According to Figure 4d, there seems no convincing significance of MIC between the enzyme-modified bacteria and bacteria with free enzyme.

Figure 4d clearly shows that the enzyme-modified bacteria are much more protected than the bacteria exposed to free enzyme. We don't understand the source of the reviewer's confusion.

10. All the figures, including the schematic figure, would be better to be improved for better presentation.

We think the figures are very clear. Everyone who has read the paper or seen the figures in presentations agrees.

11. The authors declaim that this method enables new possibilities for microbial design in biotechnology, medicine, and environmental applications where genetic modification is impractical or undesirable. Thus, more evaluations should be performed to solid their future applications.

This comment is vague. In general, assessing specific applications is the subject of follow up work.

12. The authors measured optical density (OD) in numerous experiments but failed to specify the wavelength employed in these measurements. For clarity and reproducibility, this essential detail should be reported.

The standard to measure bacterial concentrations is OD₆₀₀. We mention this in Methods and replaced the mentions in the paper with OD₆₀₀.

13. Check and correct the typos in the manuscript (such as Line545, 571, 665, legends and caption in Supplementary Figure 3, like Ecoli, C, 40μM + 27C?)

Corrected.

11th Dec 2025

Manuscript Number: MSB-2025-13416-T

Title: A universal surface functionalization technique to chemically enhance live microbial cells

Dear Prof Cordero,

Thank you for the submission of your revised manuscript to Molecular Systems Biology. Although the previous Reviewer 1 is now fully supportive of your manuscript, Reviewer 3 did not find the revisions sufficiently addressed their previous concerns, and therefore we involved a trusted advisor for arbitration. In general they found that the topic is of broad interest for the field and that the method could potentially be a strong addition to the microbiome space. They also found your explanation on the focus and distinction of your method compared to other similar methods mentioned by the reviewers sufficient and was satisfied in terms of the potential for application. However, they also mentioned that one aspect that lowered their enthusiasm was that applications outside of simple lab cultures were not demonstrated, but that nevertheless they would support publication in Molecular Systems Biology. Therefore I am pleased to inform you that we will be able to accept your manuscript pending the following final amendments and response to the remaining suggestions by the arbitrator given below.

1) Please provide the manuscript as a .docx or LaTeX file with no tracked changes.

2) Please indicate the corresponding author(s) with a symbol in the author list and provide the email address(es) on the title page of the manuscript.

3) Please download the EMBO Press "Author Checklist" and complete all relevant questions. This file should be uploaded with your submission. This file can be downloaded from our website at:

<https://www.embopress.org/page/journal/17444292/authorguide>

4) In the main manuscript file, please include keywords to max. 5.

5) Please include a Data availability section placed after the Methods section describing how the data, code etc. have been made available. This section needs to be formatted according to the example below.

"The datasets and computer code produced in this study are available in the following databases:

- Chip-Seq data: Gene Expression Omnibus GSE46748 (<https://www.ncbi.nlm.nih.gov/geo/query/acc.cgi?acc=GSE46748>)

- Modeling computer scripts: GitHub (<https://github.com/SysBioChalmers/GECKO/releases/tag/v1.0>)

- [data type]: [full name of the resource] [accession number/identifier] ([doi or URL or identifiers.org/DATABASE:ACCESSION])"

If your study does not include datasets, please insert the following statement: 'This study includes no data deposited in external repositories'.

6) Please rename "Conflict of Interest" to "Disclosure and competing interests statement". In addition, employment in a biotech company should be included in this statement. For patents and patent applications, disclosure of the following information is requested: patent applicant (whether author or institution), name of inventor(s), application number, status of application, specific aspect of manuscript covered in patent application.

7) Please correct the reference citation in the reference list to be alphabetical (not numerical). Where there are more than 10 authors on a paper, only the first 10 should be listed, followed by "et al.". Please check "Author Guidelines" for more information: <https://link.springer.com/journal/44320/submission-guidelines#cms-Reference-guidelines>

8) All Materials and Methods need to be described in the main text using our 'Structured Methods' format. According to this format, the Methods section includes a Reagents and Tools Table (listing key reagents, experimental models, software and relevant equipment and including their sources and relevant identifiers) followed by a Methods and Protocols section describing the methods, ideally using a step-by-step protocol format. The aim is to facilitate adoption of the methodologies across labs.

Please download and fill our Reagents and Tools Table template (.docx), which you can find in our author guidelines:

<https://www.embopress.org/doi/10.15252/msb.20178071>. "

9) In the Methods, please take care of the following:

- Please ensure that the sequence of oligos used are included in the Methods (or if included in table format, that the table is included in the Appendix), and that this is indicated in the Reagents and Tools table as well as the Author checklist.

- Please ensure that a statement on whether or not blinding was done is included in the Methods even if no blinding was done. Please also be sure to update the Author Checklist with this information and where it can be found in the manuscript.

10) Please place individual sections of the manuscript in the following order: Title page - Abstract & Keywords - Introduction - Results - Discussion - Methods - Data Availability - Acknowledgements - Disclosure and Competing Interests Statement - References - Figure Legends - Expanded View Figure Legends.

11) For the figures and figure legends, please take care of the following:

- The figures need to be removed from the main manuscript and uploaded as individual high-resolution Figure files, with the

legends remaining in the main manuscript file placed below the References.

- Please make sure to update the callouts of all figures in the main manuscript text. All figure callouts should be listed sequentially; currently a callout is missing for Supplementary Fig. 2.

12) Table 1 in the manuscript needs to be black & white and editable, otherwise it should be uploaded as an EV table with the nomenclature Table EV1.

13) Please upload the Appendix as a single PDF (no separate image files are needed). The title page should contain "Appendix for + manuscript title" and a Table of Contents with page numbers for the listed items. The nomenclature should be Appendix Figure Sx and Appendix Table Sx throughout the manuscript and Appendix PDF.

14) Synopsis:

- Synopsis image: Please provide a graphic that summarises the main findings of the manuscript on a glance and upload it as a high-resolution jpeg file 550 pixels wide x (300-600) pixels high.

- Synopsis text: Please provide a separate word document including a short standfirst (maximum of 300 characters, including spaces) and up to 5 bullet points to summarise the key NEW findings. They should be designed to be complementary to the abstract - i.e. not repeat the same text. We encourage inclusion of key acronyms and quantitative information (maximum of 30 words / bullet point). Please use the passive voice.

15) Instructions for providing Source Data will be sent to you in a separate email. Please ensure that a completed Source Data checklist is uploaded as a Related Manuscript File when submitting your revised manuscript. Source Data should be organized as a single source data file (zipped) per figure for main figures (all EV and/or Appendix figure Source Data can be included in a single folder), with the panels clearly visible in the folder structure instead of a single excel file for all Source Data. e.g. all the Source data files for figure 1 need to be saved in a single folder and this needs to be zipped and then uploaded as "SD figure 1.zip" file.

16) In a routine figure check, we found that Figure 5H contains a white line in the background information, which may only be a conversion error, but we would like to confirm whether this could indicate a splice or a removal of information. Please be sure to provide source data for this figure so that we can determine the source of this white line.

17) As part of the EMBO Publications transparent editorial process initiative (see our policy here:

https://www.embopress.org/transparent-process#Review_Process), Molecular Systems Biology will publish online a Peer Review File (PRF) to accompany accepted manuscripts. This file will be published in conjunction with your paper and will include the anonymous referee reports, your point-by-point response and all pertinent correspondence relating to the manuscript. Let us know whether you agree with the publication of the PRF and as here, if you want to remove or not any figures from it prior to publication. Please note that the Authors checklist will be published at the end of the PRF.

18) After your paper is published, we may promote it on social media. If you have any handles or hashtags for Bluesky you would like included, please let us know.

19) Please provide a point-by-point letter INCLUDING my comments as well as the remaining reviewer/arbitrator comments and your detailed responses (as Word file).

I look forward to reading a new revised version of your manuscript as soon as possible.

Yours sincerely,

Poonam Bheda, PhD
Scientific Editor
Molecular Systems Biology

Reviewer #1:

The authors have gone through multiple rounds of revisions - they've done an excellent job at every stage and in addressing my remaining comments. I have no further reservations

Reviewer #3:

I have carefully read the authors' point-by-point responses to the reviewers' comments and the revised manuscript. The authors have attempted to clarify certain aspects, yet the fundamental issues remain unresolved, such as the critical questions about the technical innovation beyond existing literature, the quantitative validation of the method, the direct comparison with established techniques, and its general applicability. Without addressing these concerns from the reviews, it's hard to consider acceptance for publication.

Remaining suggestions from the arbitrating reviewer:

1. I appreciate that it's very hard to estimate functional requirements with strain, environment and function dependent growth and labelling requirements. However, I wonder if a couple of sentences in discussion referencing measured growth rates for strains in complex settings, thinking about applications where this would be suited and estimating how many divisions would be feasible/accommodated for functional retention in a complex environment might help.
2. I would also insist that all data points are shown in all graphs so that the underlying variation is evident. In some instances variability is quite high and only 3 replicates tested across most experiments.

Point-by-point

- 1) Please provide the manuscript as a .docx or LaTeX file with no tracked changes.

DONE

- 2) Please indicate the corresponding author(s) with a symbol in the author list and provide the email address(es) on the title page of the manuscript.

DONE

- 3) Please download the EMBO Press "Author Checklist" and complete all relevant questions. This file should be uploaded with your submission. This file can be downloaded from our website at: <https://www.embopress.org/page/journal/17444292/authorguide>

DONE

- 4) In the main manuscript file, please include keywords to max. 5.

DONE

5) Please include a Data availability section placed after the Methods section describing how the data, code etc. have been made available. This section needs to be formatted according to the example below.

"The datasets and computer code produced in this study are available in the following databases:

- Chip-Seq data: Gene Expression Omnibus GSE46748
(<https://www.ncbi.nlm.nih.gov/geo/query/acc.cgi?acc=GSE46748>)

- Modeling computer scripts: GitHub
(<https://github.com/SysBioChalmers/GECKO/releases/tag/v1.0>)

- [data type]: [full name of the resource] [accession number/identifier] ([doi or URL or identifiers.org/DATABASE:ACCESSION])"

If your study does not include datasets, please insert the following statement: 'This study includes no data deposited in external repositories'.

NA

6) Please rename "Conflict of Interest" to "Disclosure and competing interests statement". In addition, employment in a biotech company should be included in this statement. For patents and patent applications, disclosure of the following information is requested: patent applicant (whether author or institution), name of inventor(s), application number, status of application, specific aspect of manuscript covered in patent application.

DONE

7) Please correct the reference citation in the reference list to be alphabetical (not numerical). Where there are more than 10 authors on a paper, only the first 10 should be listed, followed by "et al.". Please check "Author Guidelines" for more information: <https://link.springer.com/journal/44320/submission-guidelines#cms-Reference-guidelines>

DONE

8) All Materials and Methods need to be described in the main text using our 'Structured Methods' format. According to this format, the Methods section includes a Reagents and Tools Table (listing key reagents, experimental models, software and relevant equipment and including their sources and relevant identifiers) followed by a Methods and Protocols section describing the methods, ideally using a step-by-step protocol format. The aim is to facilitate adoption of the methodologies across labs.

DONE

Please download and fill our Reagents and Tools Table template (.docx), which you can find in our author guidelines: <https://www.embopress.org/page/journal/14693178/authorguide#structuredmethods>.

DONE

An example of a Method paper with Structured Methods can be found here: <https://www.embopress.org/doi/10.15252/msb.20178071>. "

9) In the Methods, please take care of the following:

- Please ensure that the sequence of oligos used are included in the Methods (or if included in table format, that the table is included in the Appendix), and that this is indicated in the Reagents and Tools table as well as the Author checklist.

DONE

- Please ensure that a statement on whether or not blinding was done is included in the Methods even if no blinding was done. Please also be sure to update the Author Checklist with this information and where it can be found in the manuscript.

DONE

10) Please place individual sections of the manuscript in the following order: Title page - Abstract & Keywords - Introduction - Results - Discussion - Methods - Data Availability - Acknowledgements - Disclosure and Competing Interests Statement - References - Figure Legends - Expanded View Figure Legends.

DONE

11) For the figures and figure legends, please take care of the following:
- The figures need to be removed from the main manuscript and uploaded as individual high-resolution Figure files, with the legends remaining in the main manuscript file placed below the References.

DONE

- Please make sure to update the callouts of all figures in the main manuscript text. All figure callouts should be listed sequentially; currently a callout is missing for Supplementary Fig. 2.

DONE

12) Table 1 in the manuscript needs to be black & white and editable, otherwise it should be uploaded as an EV table with the nomenclature Table EV1.

DONE

13) Please upload the Appendix as a single PDF (no separate image files are needed). The title page should contain "Appendix for + manuscript title" and a Table of Contents with page numbers for the listed items. The nomenclature should be Appendix Figure Sx and Appendix Table Sx throughout the manuscript and Appendix PDF.

DONE

14) Synopsis:
- Synopsis image: Please provide a graphic that summarises the main findings of the manuscript on a glance and upload it as a high-resolution jpeg file 550 pixels wide x (300-600) pixels high.

DONE

- Synopsis text: Please provide a separate word document including a short standfirst (maximum of 300 characters, including spaces) and up to 5 bullet points to summarise the key NEW findings. They should be designed to be complementary to the abstract -

i.e. not repeat the same text. We encourage inclusion of key acronyms and quantitative information (maximum of 30 words / bullet point). Please use the passive voice.

DONE

DONE

15) Instructions for providing Source Data will be sent to you in a separate email. Please ensure that a completed Source Data checklist is uploaded as a Related Manuscript File when submitting your revised manuscript. Source Data should be organized as a single source data file (zipped) per figure for main figures (all EV and/or Appendix figure Source Data can be included in a single folder), with the panels clearly visible in the folder structure instead of a single excel file for all Source Data. e.g. all the Source data files for figure 1 need to be saved in a single folder and this needs to be zipped and then uploaded as "SD figure 1.zip" file.

We have this file ready. It's large, though (a few hundred MBs all together)

16) In a routine figure check, we found that Figure 5H contains a white line in the background information, which may only be a conversion error, but we would like to confirm whether this could indicate a splice or a removal of information. Please be sure to provide source data for this figure so that we can determine the source of this white line.

DONE

17) As part of the EMBO Publications transparent editorial process initiative (see our policy here: https://www.embopress.org/transparent-process#Review_Process), Molecular Systems Biology will publish online a Peer Review File (PRF) to accompany accepted manuscripts. This file will be published in conjunction with your paper and will include the anonymous referee reports, your point-by-point response and all pertinent correspondence relating to the manuscript. Let us know whether you agree with the publication of the PRF and as here, if you want to remove or not any figures from it prior to publication. Please note that the Authors checklist will be published at the end of the PRF.

DONE

18) After your paper is published, we may promote it on social media. If you have any handles or hashtags for Bluesky you would like included, please let us know.

DONE

19) Please provide a point-by-point letter INCLUDING my comments as well as the remaining reviewer/arbitrator comments and your detailed responses (as Word file).

DONE

29th Jan 2026

Manuscript Number: MSB-2025-13416R

Title: A universal surface functionalization technique to chemically enhance live microbial cells

Dear Prof Cordero,

In checking your manuscript and point-by-point response for the remaining issues, we note that not all requests had been fully addressed. Please address the following and resubmit your manuscript when all items have been completed:

Please provide a point-by-point response to the remaining reviewer/arbitrator comments.

Please provide the full Source Data as instructed, including the Source Data checklist that should be uploaded as a Related Manuscript file. Source Data should be organized as a single source data file (zipped) per figure for main figures (all EV and/or Appendix figure Source Data can be included in a single folder), with the panels clearly visible in the folder structure instead of a single excel file for all Source Data. e.g. all the Source data files for figure 1 need to be saved in a single folder and this needs to be zipped and then uploaded as "SD figure 1.zip" file.

In particular, please be sure to provide the Source Data for Figure 5H that contains a white line in the background information so that we can determine whether this was a conversion error or this could indicate a splice or a removal of information.

In addition, please address these remaining formatting issues in the manuscript:

- 'Materials and Methods' should be labeled as 'Methods'
- Table 1 needs to be placed after the Figure legends
- callouts for Figure EV2 and Table EV1 missing in the manuscript
- Table EV1 (that was previously included in Appendix) needs to be removed from manuscript and uploaded as a separate file

If you have any questions, please contact me or our editorial office.

Yours sincerely,

Poonam Bheda, PhD
Senior Scientific Editor
Molecular Systems Biology

Reviewer #1:

The authors have gone through multiple rounds of revisions - they've done an excellent job at every stage and in addressing my remaining comments. I have no further reservations

Thank you for your positive assessment.

Reviewer #3:

I have carefully read the authors' point-by-point responses to the reviewers' comments and the revised manuscript. The authors have attempted to clarify certain aspects, yet the fundamental issues remain unresolved, such as the critical questions about the technical innovation beyond existing literature, the quantitative validation of the method, the direct comparison with established techniques, and its general applicability. Without addressing these concerns from the reviews, it's hard to consider acceptance for publication.

We have provided clear evidence in the manuscript demonstrating the technical advances of our method, including direct, head-to-head comparisons with existing approaches. We also demonstrate multiple potential applications of the technique, while recognizing that detailed, problem-specific applications are the subject of future work. Importantly, the method has been quantitatively validated well beyond the current standards of the field. Taken together, we feel that these criticisms do not fairly reflect the scope or rigor of the work presented.

Remaining suggestions from the arbitrating reviewer:

1. I appreciate that it's very hard to estimate functional requirements with strain, environment and function dependent growth and labelling requirements. However, I wonder if a couple of sentences in discussion referencing measured growth rates for strains in complex settings, thinking about applications where this would be suited and estimating how many divisions would be feasible/ accommodated for functional retention in a complex environment might help.

This is an interesting idea; however, growth rates for strains in natural settings have not been reliably quantified. Moreover, we would argue that the relevant timescale is generational rather than absolute time (hours). As a result, speculation about phenotypes and growth rates would likely weaken, rather than strengthen, the paper.

2. I would also insist that all data points are shown in all graphs so that the underlying variation is evident. In some instances variability is quite high and only 3 replicates tested across most experiments.

All graphs follow strict guidelines for data presentation. I have verified that the error bars in all figures are small. In cases where individual data points deviate substantially from the error bars, these are explicitly shown in the plots (e.g., Fig 4). In addition, Fig. S3 in the Supplementary Information presents the full individual data traces for replicate experiments for which only summary statistics are shown in the main text, thereby providing complete transparency of the underlying data.

11th Feb 2026

Manuscript number: MSB-2025-13416RR

Title: A universal surface functionalization technique to chemically enhance live microbial cells

Dear Prof Cordero,

Thank you again for sending us your revised manuscript. We are now satisfied with the modifications made and I am pleased to inform you that your paper has been accepted for publication.

You may qualify for financial assistance for your publication charges - either via a Springer Nature fully open access agreement or an EMBO initiative. Check your eligibility: <https://link.springer.com/journal/44320/how-to-publish-with-us>

Yours sincerely,

Sincerely,

Poonam Bheda, PhD
Scientific Editor
Molecular Systems Biology

>>> Please note that it is Molecular Systems Biology policy for the transcript of the editorial process (containing referee reports and your response letter) to be published as an online supplement to each paper. If you do NOT want this, you will need to inform the Editorial Office via email immediately. More information is available here: <https://link.springer.com/partners/embo-press/editorial-policies#Peer%20review>